# Echo-GAT: Debiasing Graph Attention with Echo Nodes and Degree Diversity for Heterophilic Graphs

## Abstract

Attention mechanisms have become a de facto standard for enhancing the expressivity of deep learning models, achieving remarkable success in graph data. Recent studies have shown that attention-based graph neural networks (GNNs) often perform poorly on heterophilic graphs and have attributed this degradation primarily to low levels of homophily. In contrast to this prevailing explanation, we find that on heterophilic graphs, under standard graph attention mechanisms, node-level homophily shows only a weak correlation with prediction accuracy, and nodes with lower homophily ratios can even achieve higher accuracy on average. These observations suggest that homophily alone is insufficient to explain the failure of graph attention. In this work, we show that standard graph attention networks exhibit a systematic performance imbalance across nodes with different degrees of diversity, favoring structurally inhomogeneous nodes (i.e., those with significantly divergent degrees compared to their neighbors). To mitigate this bias, we propose a graph attention optimization framework that integrates augmented feature attention and degree diversity-aware attention score to mitigate node-level structural bias. Experiments show that the proposed method consistently outperforms strong GAT variants and state-of-the-art heterophily-oriented GNNs. Moreover, it maintains stable performance gains across nodes with varying heterophily levels, demonstrating its effectiveness on diverse graph structures.

## 1 Introduction

Since the seminal work by Velickovic et al. (Velickovic et al., 2018), attention mechanism has spread over many applications of deep neural networks. In particular, the attention that is extended to accommodate irregular graph-structured data, known as graph attention, shows promising performance on node-level and graph-level prediction tasks (Liu et al., 2020; Bo et al., 2021). In recent years, some variants of the vanilla graph attention networks (GAT (Velickovic et al., 2018)) have been proposed to improve the performance, such as GATv2 (Brody et al., 2022) and AERO-GNN (Lee et al., 2023). Despite their success, graph attention models still face notable challenges when applied to heterophilic graphs, where connected nodes often belong to different classes. Due to the widespread presence of heterophilic graphs in real-world applications (e.g., web-page networks, biological networks, and recommender systems), this issue has emerged as an active research topic in graph neural networks. Most existing studies attribute the performance degradation of graph attention networks on heterophilic graphs to low homophily, arguing that message passing becomes less effective when connected nodes tend to belong to different classes. Following this explanation, a large body of work focuses on identifying homophilic and heterophilic edges or reconstructing node–node interactions to improve attention-based message passing (Wang et al., 2024b; Jiang et al., 2024). For example, SA-GNN (Huang et al., 2023b) learns a binary classifier to distinguish homophilic and heterophilic edges, and subsequently prunes heterophilic edges to increase the overall homophily ratio.

However, in this work, we reveal a contrasting phenomenon: on heterophilic graphs, the influence of homophily on graph attention is far from uniform at the node level, as illustrated in the first column of Fig. 1. To ensure that this observation is statistically reliable, all results in Fig. 1 are averaged over 10 independent runs with different random seeds, and the error bars denote the standard deviation. In particular, node-level homophily shows a weak correlation with prediction accuracy, and nodes with lower homophily ratios can

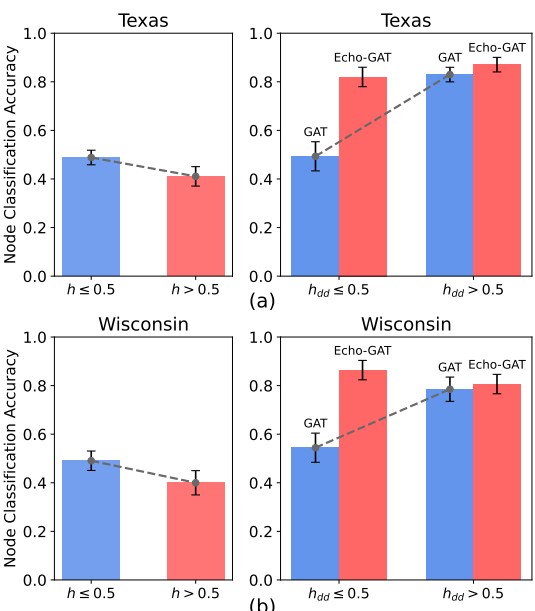

Figure 1: Comparison of node classification accuracy on the Texas and Wisconsin datasets. Left: node accuracy of GAT under different homophily levels $h$ (homophily level). Right: node accuracy under different degree diversity levels $h_{dd}$ (degree diversity), comparing GAT with the proposed method (i.e., Echo-GAT).

even achieve higher accuracy on average. These observations indicate that homophily alone is insufficient to characterize node-level behavior in graph attention networks.

Homophily measures label-level similarity among neighboring nodes, whereas degree diversity reflects structural contrast within a node's neighborhood; these two properties are conceptually independent. Our empirical observations show that, on heterophilic graphs, degree diversity correlates more strongly with node-level attention performance than homophily.

This finding poses an additional challenge for designing effective graph attention mechanisms on heterophilic graphs. When homophily fails to reliably characterize node-level behavior, attention models can no longer depend on label or feature similarity as a stable cue for neighbor selection. From the perspective of attention design, this challenge is particularly pronounced on heterophilic graphs, where node neighborhoods often exhibit heterogeneous degree patterns, offering rich structural cues for distinguishing neighbors. However, when such degree heterogeneity is insufficient (i.e., nodes exhibit low degree diversity), their neighborhoods become structurally uniform, with neighbors sharing similar degrees. In this case, the lack of structural differentiation makes it difficult for the attention mechanism to effectively distinguish among neighbors, causing attention weights to be distributed in a biased or uninformative manner. As a result, message aggregation becomes less discriminative, leading to degraded prediction performance. In contrast, higher degree diversity introduces richer structural variation, enabling attention mechanisms to better differentiate neighbors and perform more effective message passing.

Motivated by the above observations, we focus on degree diversity as a key structural factor influencing graph attention behavior at the node level. Empirically, we observe from the second column of Fig. 1 that node classification accuracy under GAT exhibits a clear dependency on degree diversity: nodes with low degree diversity consistently achieve lower accuracy, while those with higher degree diversity attain significantly better performance. Compared with grouping by homophily, the accuracy curves under degree diversity grouping exhibit a much clearer separation between different structural regimes, indicating that degree diversity provides a stronger structural signal for explaining node-level performance variations. This pronounced separation indicates that degree diversity provides a strong and discriminative structural signal for characterizing node-level behavior in graph attention networks.

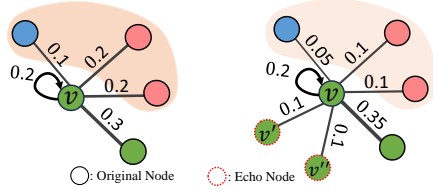

Figure 2: Node attention changes between the original graph and the extended graph.

In this paper, we present a method to adjust the bias of graph attention on heterophilic graphs. The core idea is the introduction of echo nodes, the copies of the central nodes. The effect of this operation is illustrated in Fig. 2, where node $v$ attends its neighbors with different weights and colors denote labels. Once the echo nodes are added to the neighborhood of node $v$, the local neighborhood structure is regularized, leading to increased degree diversity and more balanced structural interactions during attention aggregation.

Then we design an extended attention for the augmented graph, which consists of two parts. The first part is feature-level attention, which jointly aggregates representations from the original nodes and their corresponding echo nodes. The second part is a degree diversity-aware attention score, which provides the ability to account for degree disparities at both local and high-order scales. As a result, the proposed attention mechanism can more effectively capture node interactions in structurally inhomogeneous neighborhoods.

We summarize our contributions as follows:

- We propose a novel method to extend GAT to heterophilic graphs by introducing echo nodes, which dilute the attention weights of the central node toward its heterophilic neighbors.

- We further exploit local degree diversity among connected nodes to refine the relative importance measurement between the central node and its neighbors, implemented through a combination of neural network learning and edge ranking.

- Experiments on graphs spanning the full heterophily spectrum demonstrate the superiority of our method over both GAT variants and heterophily-oriented GNNs, regardless of whether they employ attention mechanisms.

## 2 Related Work

Most GNNs employ the message passing (MP) to facilitate the feature propagation among nodes and their neighbors, where prominent examples include GCN (Kipf & Welling, 2017) and GAT (Velickovic et al., 2018). (Wu et al., 2019) further decouple the MP to explore the global structural information. Unfortunately, the homophily implicit in MP limits the generalization of GNNs to heterophilic graphs, which is outlined in (Sun et al., 2024; Ai et al., 2024; Huang et al., 2023a; 2025). They show that unlike "like attracts like" in homophilic graphs, different nodes in heterophilic graphs tend to be linked. In this case, GNNs may struggle to perform well. Recent works have begun to revisit the heterophily by designing wider messaging ranges to capture distant but similar nodes in the heterophilic graph (Song et al., 2023; Huang et al., 2024). JKNet and Mix-Hop (Xu et al., 2018; Abu-El-Haija et al., 2019) transform and connect multilayer neighbor representations, while DAGNN and GPRGNN (Liu et al., 2020; Chien et al., 2021) use graph diffusion to capture higher-order neighbors in heterophilic graphs. Other studies such as graph contrastive learning (Wang et al., 2024a) and graph structure learning (Battiloro et al., 2024) have shown that can improve GNNs on heterophilic graphs.

Attention-based GNNs aim to learn to propagate by inferring the relational importance between node pairs. Among many, GAT and its variants (Velickovic et al., 2018; Brody et al., 2022; Lee et al., 2023) learn edge attention, aiming at inferring the importance, or weight, of each neighbor w.r.t. central node. However, it remains a big challenge to apply graph attention mechanism to heterophilic graphs (Xu et al., 2025; Pan et al., 2024). Current approaches towards this issue mainly put efforts into identifying heterophilic edges before performing graph attention and convolution. For example, WRGAT (Suresh et al., 2021)

converts heterophily graphs into multi-relational graphs, where the proximity and structural information are characterized and used to craft different types of edges, followed by the vanilla graph attention operations. DIR-GAT (Rossi et al., 2023) makes use of the edge direction to indicate the edge type. Those solutions demonstrate the effectiveness in improving the overall node classification performance of GAT. Different from the existing graph attention based GNNs (Kazi et al., 2023; de Ocáriz Borde et al., 2023), our model focuses on addressing the fairness of graph attention mechanism w.r.t the local degree diversity of a graph, thus improving the overall performance of GNNs on a wide range of graph data.

## 3 Notations and Preliminaries

This section introduces the preliminaries of our work, including notations and definitions.

### 3.1 Notations

Let $\mathcal{G} = (\mathbf{A}, \mathbf{E}, \mathbf{X}, \mathbf{Y})$ represent a graph with node features and labels, where $\mathbf{A} \in \{0, 1\}^{n \times n}$ denotes the adjacency matrix, $\mathbf{E}$ is the edge set, and $\mathbf{X} \in \mathbb{R}^{n \times d}$ is the node feature matrix. For nodes $i, j \in \{1, 2, ..., n\}$, $\mathbf{A}_{i,j} = 1$ if and only if nodes $i$ and $j$ are connected by an edge in $\mathcal{G}$, and $\mathbf{X}_i \in \mathbb{R}^d$ represents the features of node $i$. $\mathbf{X}$ denotes the set of ground-truth node labels. The set of neighbors of the central node $i$ is denoted by $\mathcal{N}(i) = \{j \mid \mathbf{A}_{i,j} = 1\}$. $\mathbf{D} \in \mathbb{R}^{n \times n}$ represents the degree matrix of graph $\mathcal{G}$, which is a diagonal matrix, where the $i$-th diagonal element $\mathbf{D}_{i,i} = d_i = |\mathcal{N}(i)|$ represents the degree of node $i$.

In this paper, we primarily focus on the node classification task (Chien et al., 2021; Ma et al., 2022), which is one of the most important machine learning tasks on graphs. Node classification is a semi-supervised learning problem defined as follows:

**Definition 1 (Node Classification)** *Node classification is a task of learning the conditional probability* $Q(\mathbf{Y} \mid \mathcal{G}; \Theta)$ *to distinguish the class of each unlabeled node in a single graph* $\mathcal{G} = \left(\mathbf{A}, \mathbf{X}, \hat{\mathbf{Y}}\right)$, *where* $\Theta$ *is the model parameters and* $\hat{\mathbf{Y}}|$ *is the partially known set of node labels.*

### 3.2 Node Homophily

Homophily is an inherent property of graphs, indicating that connected node pairs tend to be similar. In the context of node classification, this implies a higher probability that connected nodes share the same label. In this work, we focus on the homophily of the local field around central node. Therefore, we adopt the definition of node-level homophily used in the prior works (Zhu et al., 2020; Lim et al., 2021), which reads as

**Definition 2 (Node Homophily Ratio (Zhu et al., 2020))** *Given the central node* $i$ *in the graph* $\mathcal{G} = (\mathbf{A}, \mathbf{X}, \mathbf{Y})$, *the node homophily ratio is defined as the proportion of edges connecting node* $i$ *to another node* $j$ *with the same label. Formally, this metric reads as*

$$h_i = \frac{\sum_{j \in \mathcal{N}(i)} \mathbb{I}(y_i = y_j)}{|\mathcal{N}(i)|}, \tag{1}$$

*where* $\mathbb{I}(\cdot)$ *is the indicator function (i.e., when the condition* $\cdot$ *holds,* $\mathbb{I}(\cdot) = 1$; *otherwise,* $\mathbb{I}(\cdot) = 0$).

Accordingly, the node homophily $h_i \to 1$ implies the strong homophily of node $i$'s neighborhood, while when $h_i \to 0$, the node is considered to have strong heterophily.

### 3.3 Degree Diversity

Next, we characterize the structural connection patterns. While homogeneity, a property that characterizes the evenness of degree distribution, is commonly measured at the graph level (Li et al., 2022; Wang et al., 2022), such metrics often overlook structural variations at the local level. To capture local connectivity properties, we propose the node-wise degree diversity ratio, a metric that mainly relies on structural characteristics to reflect the diversity of local connectivity patterns.

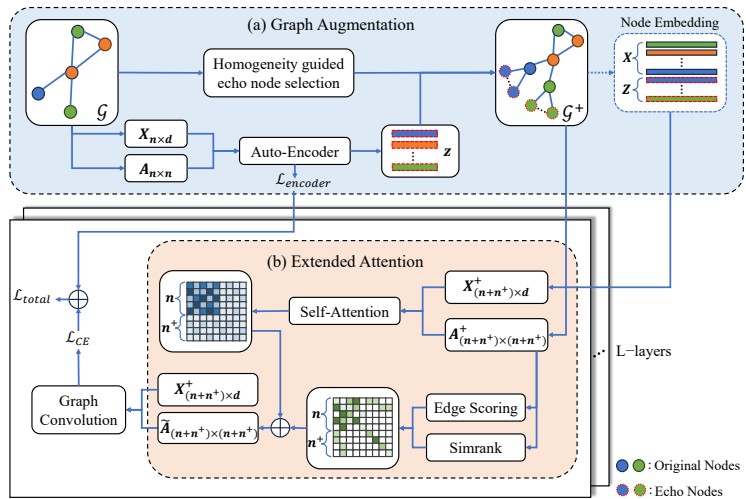

Figure 3: The flowchart of the proposed Echo-GAT. Specifically, given the heterophilic graph $\mathbf{A}$ and the corresponding features $\mathbf{X}$, Echo-GAT first employs the Variational Graph Auto-Encoder (VGAE) to generate echo nodes for nodes with low degree diversity, resulting in an augmented graph (i.e., subfigure (a)). After that, Echo-GAT introduces the extended attention for the augmented graph to capture the feature-level and degree diversity-aware information at both local and high-order levels (i.e., subfigure (b)). Finally, the graph supervision loss and the VGAE reconstruction loss are jointly optimized, enabling the learning of informative echo node representations and degree-aware attention weights.

Unlike statistics such as the variance of neighbor degrees, which measure the dispersion among neighbors themselves, our formulation focuses on the degree deviation between the central node and its local neighborhood. This node-wise perspective is important for our method design, as it provides a structural signal for identifying nodes whose connectivity patterns deviate from their surrounding context.

The formal definition is as follows:

**Definition 3 (Degree Diversity Ratio)** *Given a central node $i$ in the graph $\mathcal{G} = (\mathbf{A}, \mathbf{X}, \mathbf{Y})$, the degree diversity ratio measures the deviation between the degree of the central node and the average degree of its neighbors, formulated as*

$$\mathbf{h}_{dd}(i) = Norm\left(\left|d_i - \frac{1}{|\mathcal{N}(i)|}\sum_{j \in \mathcal{N}(i)} d_j\right|\right), \tag{2}$$

*where $|\cdot|$ denotes the absolute value. The function $Norm(\cdot)$ denotes min–max normalization defined as*

$$Norm(x_i) = \frac{x_i - x_{\min}}{x_{\max} - x_{\min}}, \tag{3}$$

*where $x_{\min}$ and $x_{\max}$ represent the minimum and maximum values computed over all nodes in the graph.*

Node degree diversity measures the extent to which a central node has a similar degree to those of its neighbors. A lower node degree-diversity ratio $\mathbf{h}_{dd}(i)$ indicates a smaller degree difference between the central node and its neighbors, implying a more structurally balanced local neighborhood. In contrast, a higher $\mathbf{h}_{dd}(i)$ indicates that the degree of the central node significantly differs from that of its neighbors, suggesting a stronger structural mismatch in the local connectivity pattern.

## 4  Methodology

**Overview.** To rectify the node-level bias of graph attention induced by low degree diversity, we propose a graph attention optimization framework that explicitly incorporates degree-based structural information

into message aggregation, as illustrated in Fig. 3. Our model consists of two components: (i) **Graph augmentation via echo nodes.** We employ a variational graph autoencoder (VGAE) to generate echo nodes for selected central nodes, augmenting the original graph structure. This augmentation regularizes local neighborhood structures by supplementing the neighborhoods of nodes with insufficient degree diversity, thereby increasing overall degree diversity. (ii) **Extended attention on the augmented graph.** On the augmented graph, we design an extended attention mechanism that jointly incorporates feature-level signals and degree-based structural information at both local and high-order levels. Specifically, the attention scores derived from different structural and semantic sources are integrated into a unified attention weight, enabling message passing in graph attention to encode richer feature representations together with degree diversity information at multiple structural scales.

## 4.1 Graph Augmentation

Based on our earlier analysis, we attribute the node-level performance bias of graph attention to structural imbalance in local neighborhoods, which is closely related to insufficient degree diversity. Towards this end, we extend the neighborhood of the nodes with low degree diversity. Specifically, we introduce *echo nodes*, which are virtual neighbors of the central nodes sharing similar feature distribution, as illustrated in the graph augmentation module of Fig. 3. These echo nodes reshape local neighborhood compositions and facilitate degree diversity.

When selecting nodes for neighborhood extension according to Eq. 2, we focus on nodes with both low degree diversity and low degrees (i.e., $\{i \mid d_i < \frac{1}{|\mathcal{N}(i)|} \sum_{j \in \mathcal{N}(i)} d_j\}$). This design is motivated by the observation that, for high-degree nodes, attention weights over neighbors tend to be more evenly distributed due to the normalization effect of the softmax operation, resulting in limited sensitivity to additional neighbors. Consequently, extending the neighborhoods of high-degree nodes is unlikely to substantially alter their attention distributions, whereas nodes with low degrees can benefit more from neighborhood augmentation.

**Echo Node Initialization.** Let $\mathcal{G}^+ = (\mathbf{A}^+, \mathbf{X}^+, \mathbf{V}^+)$ be the augmented graph, where $\mathbf{V}^+ = \mathbf{V}_0 \cup \mathbf{V}_e$ consists of the original node set $\mathbf{V}_0$ and the disjoint echo node set $\mathbf{V}_e$, $\mathbf{A}^+ \in \{0,1\}^{|\mathbf{V}^+| \times |\mathbf{V}^+|}$, $\mathbf{X}^+ \in \mathbb{R}^{|\mathbf{V}^+| \times d}$. For the nodes with $d_i < \frac{1}{|\mathcal{N}(i)|} \sum_{j \in \mathcal{N}(i)} d_j$ and $\mathbf{h}_{dd} > \beta$, where $\beta$ is the degree diversity threshold, we add $k$ echo nodes as the immediate neighbors for each of them, i.e., $\mathbf{A}_{i,j}^+ = \mathbf{A}_{j,i}^+ = 1$ for the augmented graph $G^+$, where $i \in V_0$ is the original node and $j \in \mathbf{V}_e$ is its echo. Moreover, there are no connections between echo nodes.

Next, we obtain node embeddings using a graph autoencoder and use them to initialize the corresponding echo nodes. Various graph autoencoder methods can be used for this purpose, such as VGAE (Kipf & Welling, 2016), GraphMAE (Hou et al., 2022), and GraphMAE2 (Hou et al., 2023). In this work, we employ VGAE to generate node embeddings. Briefly, for a given graph $\mathcal{G} = (A, X)$, VGAE encodes each node $i$ into an embedding vector $\mathbf{z}_i$. The entire process consists of two modules:

**Encoder.** The encoder infers the latent distribution of each node from the graph structure and node features. The embedding vector $\mathbf{z}_i \in \mathbb{R}^d$ of node $i$ follows a probability distribution whose mean and variance are computed using a Graph Convolutional Network (GCN). This is formulated as

$$q(\mathbf{z}_i | \mathbf{X}, \mathbf{A}) = \mathbb{N}(\mathbf{z}_i; \bar{\mu}_i, \mathrm{diag}(\bar{\sigma}_i^2)), \tag{4}$$

where $q(\cdot)$ is the approximate posterior distribution, $\mathbb{N}(\cdot)$ denotes a multivariate normal distribution, $\bar{\mu}_i$ and $\bar{\sigma}_i^2$ denote the mean and variance of the latent distribution for node $i$, respectively.

**Decoder.** The decoder reconstructs the adjacency matrix of the graph from the node embedding vectors by determining the existence of an edge between nodes $i$ and node $j$ using their embedding vectors $\mathbf{z}_i$ and $\mathbf{z_j}$. The reconstruction is performed via the inner product

$$\hat{\mathbf{A}}_{ij} = \sigma(\mathbf{z}_i^T \mathbf{z_j}), \tag{5}$$

where $\hat{\mathbf{A}}_{ij}$ is the reconstructed adjacency matrix element, and $\sigma$ is an activation function mapping values to the range [0,1].

The latent embeddings are projected to the original feature dimension before being used as echo node features, ensuring dimensional consistency with the input feature space. The learned embedding vector $\mathbf{z}_i$ is then set as the initial feature of each echo node of the target node $i$. Let $k$ copies of the embedding $\mathbf{z}_i$ be the set $\mathbf{Z}_i = \{\mathbf{z}_{i,0}, \mathbf{z}_{i,1}, \ldots, \mathbf{z}_{i,k}\}$. Note that all $k$ echo nodes are connected to the corresponding original node $i$, forming new edges, resulting in the extended adjacency matrix $A^+$. Each echo node is connected only to its corresponding original node and does not connect to any other original nodes or other echo nodes. Formally, we define the extended graph as

$$\mathbf{X}^+ = \left[\mathbf{X}, \left[\mathbf{Z}_i \mid \mathbf{h}_{dd}(i) > \beta\right]\right], \quad i = 0, \ldots, n, \tag{6}$$

$$\mathbf{A}^+ = \mathbf{A} + \sum_{i=0}^{n} \sum_{j=0}^{k} \left\{ \left(I_{i,(n+j \cdot n^+)} + I_{(n+j \cdot n^+),i}\right) \mid \mathbf{h}_{dd}(i) > \beta \right\}, \tag{7}$$

$$n^+ = \sum_{i=0}^{n} \mathbb{I}(\mathbf{h}_{dd}(i) > \beta), \tag{8}$$

where $X^+ \in \mathbb{R}^{(n+n^+) \times d}$ and $A^+ \in \mathbb{R}^{(n+n^+) \times (n+n^+)}$ are the extended node feature matrix and adjacency matrix, respectively. The notation $[\cdot]$ denotes matrix concatenation, and $n$ is the number of nodes in the original graph. The indicator function $\mathbb{I}(\cdot)$ returns 1 if the specified condition is met and 0 otherwise. Thus, $n^+$ represents the number of nodes satisfying $\mathbf{h}_{dd}(i) > \beta$. The parameters $k$ and $\beta$ are hyperparameters, where $k$ is the number of echo nodes associated with one original node, and $\beta$ represents the degree diversity threshold for selecting original nodes. $I_{i,(n+j \cdot n^+)}$ is a unit matrix with a 1 at position $(i, (n + j \cdot n^+))$, indicating the presence of an added edge. This operation results in an extended graph $\mathcal{G}^+ = (\mathbf{X}^+, \mathbf{A}^+)$, constructed from degree diversity $\mathbf{h}_{dd}$ and node embedding vectors $\mathbf{z}$.

## 4.2 Extended Attention on the Augmented Graph

After graph augmentation, we obtain an extended graph $\mathcal{G}^+ = (X^+, A^+)$ that contains both the original nodes and their corresponding echo nodes. To effectively perform message passing on such an augmented structure, in the feature space, we employ an extended attention mechanism that can jointly model interactions among original nodes as well as between original and echo nodes. Beyond feature similarity, we further incorporate structural information to account for node-level degree diversity. In particular, we model the influence of degree diversity between connected node pairs using a multi-layer perceptron (MLP) based on degree-related structural features. To capture higher-order structural relationships, we additionally leverage SimRank-based similarities as complementary edge weights. The detailed implementation of each component is described below.

### 4.2.1 Extended Featural Attention

We first employ graph attention mechanism to compute the attention scores for each edge $(i, j)$ in the extended graph $\mathcal{G}^+ = (X^+, A^+)$, similar to GAT, where node dependencies are determined based on their feature representations. That is, for each pair of adjacent nodes $i$ and $j$, the extended attention score $\alpha_{ij}^+$ is computed as

$$\alpha_{ij}^{+(\ell+1)} = \frac{\exp\left(\text{LeakyReLU}\left(\mathbf{a}^T[\mathbf{h}_i^{(\ell)} \parallel \mathbf{h}_j^{(\ell)}]\right)\right)}{\sum_{k \in \mathcal{N}^+(i)} \exp\left(\text{LeakyReLU}\left(\mathbf{a}^T[\mathbf{h}_i^{(\ell)} \parallel \mathbf{h}_k^{(\ell)}]\right)\right)}, \tag{9}$$

$$\mathbf{h}_i^{(0)} = \mathbf{x}_i^+ \tag{10}$$

where $\mathbf{h}_i$ and $\mathbf{h}_j$ denote the feature vectors of nodes $i$ and $j$, respectively, and $[\cdot \parallel \cdot]$ represents the concatenation operation. The learnable weight vector $\mathbf{a}$ is used to compute the attention score between node pairs. The function $\text{LeakyReLU}(\cdot)$ denotes the Leaky Rectified Linear Unit, and the exponential ensures non-negative scores. The denominator performs a softmax normalization over all neighbors $k \in \mathcal{N}^+(i)$ of node $i$ in the extended graph, including both original and echo nodes. As a result, the attention coefficient

$\alpha_{ij}^{+(\ell+1)}$ quantifies the relative importance of node $j$ when aggregating messages for node $i$. It can also be viewed as the dynamic connection weight between two nodes.

By computing the feature-level similarity for each node pair in the extended graph $\mathcal{G}^+$, we obtain the full attention score matrix $\alpha^{+(\ell)}$ in each layer.

### 4.2.2 Degree Diversity-Aware Attention Score

Homophily measures label-level similarity among neighboring nodes, whereas degree diversity captures structural contrast within a node's neighborhood; these two properties are conceptually independent. Degree diversity provides a more reliable structural signal for guiding attention under heterophily, as feature or label similarity may be misleading.

To complement the feature-based extended attention, we introduce a degree diversity-aware attention that captures node-level structural inhomogeneity from a topological perspective. This module assigns local and high-order structural weights to edges based on the degree disparity and multi-hop neighborhood overlap between connected nodes, enabling graph attention score to incorporate degree diversity information at multiple structural scales.

**Local Degree-Based Structural Weighting.** We model the impact of degree diversity on node-to-node interactions using a multi-layer perceptron (MLP). For each edge $(i,j)$, we construct an edge feature vector that encodes degree-related structural information, including the degrees of the two endpoints, $d_i$ and $d_j$, as well as their multiplicative interaction $d_i \cdot d_j$. These features capture both individual degree properties and pairwise degree relationships. For all edges in the graph, these feature vectors are arranged into an degree-aware edge feature matrix $E_f \in \mathbb{R}^{m \times 3}$:

$$\mathbf{E}_f = \sum_{i,j \in \mathbf{E}} [\mathbf{e}(i,j)], \quad \mathbf{e}(i,j) = [d_i, d_j, d_i \cdot d_j], \tag{11}$$

where $\mathbf{e}(\cdot)$ represents the edge feature vector, and $[\cdot]$ denotes matrix concatenation.

The resulting edge feature matrix $E_f$ is then fed to the MLP with two hidden layers. The MLP outputs a score vector $\mathbf{e_s} \in \mathbb{R}^{|E|}$, wherein each element corresponds to the score of an edge $(i_k, j_k)$:

$$\mathbf{e}_s = \mathrm{MLP}(\mathbf{E}_f; \Theta_{mlp}), \tag{12}$$

$$\mathbf{E}_s[i_k, j_k] = \mathbf{E}_s[j_k, i_k] = \mathbf{e}_s[k], \tag{13}$$

where $\Theta_{mlp}$ denotes the learnable parameters of the MLP. The score matrix $\mathbf{E}_s \in \mathbb{R}^{(n+n^+) \times (n+n^+)}$ measures the interaction strength between connected pairs from the perspective of the first-order structural correlation.

**High-Order Degree-Based Structural Similarity.** In order to incorporate the higher-order structural correlation in the edge scoring, we employ the ready-made SimRank (Jeh & Widom, 2002), a method based on the intuition that similar nodes are referenced by similar nodes. The structural similarity $s(i,j)$ between nodes in the graph is recursively defined as

$$s(i,j) = \frac{C}{d_i \cdot d_j} \sum_{p \in \mathcal{N}(i)} \sum_{q \in \mathcal{N}(j)} s(p,q), \tag{14}$$

$$\mathbf{E}_h = \mathrm{Norm}(\mathbf{E}_s + s(i,j)), \tag{15}$$

where $s(i,j)$ represents the SimRank similarity between nodes $i$ and $j$, and $C$ is a normalization constant. The double summation iterates over all pairs of neighboring nodes $(p,q)$, where $p \in \mathcal{N}(i)$ and $q \in \mathcal{N}(j)$, accumulating their similarity scores $s(p,q)$ between the neighboring nodes. $\mathbf{E}_s$ denotes the previously calculated edge score, while $\mathrm{Norm}(\cdot)$ scales the value to the range $[0, 1]$.

Incorporating the SimRank similarity $s(i,j)$ enriches the structural attention score by complementing local degree-based structural signals with higher-order neighborhood information. Specifically, the local structural attention score captures degree disparity between connected nodes, while SimRank accounts for their structural similarity induced by multi-hop neighborhood overlap. Together, these components mitigate node-level structural inhomogeneity and promote more consistent attention aggregation across diverse neighborhood configurations.

### 4.2.3 Attention Integration

After completing the aforementioned steps, the extended attention scores $\alpha^+$ and the structure-based edge weights $\mathbf{W}$ collectively determine the final weight matrix $\widetilde{\mathbf{A}}$ for each edge in the graph. This matrix serves as the weighted coefficient in the node convolution process. The $\widetilde{\mathbf{A}}$ is first normalized in the way

$$\widetilde{\mathbf{A}}^{(\ell)} = \text{Norm}\left(\left(\alpha^{+(\ell)} + \mathbf{E}_h\right) \odot \mathbf{A}^+\right), \tag{16}$$

where $\alpha^{+(\ell)}$ represents the extended attention score matrix, $\mathbf{E}_h$ is the edge weight matrix obtained by combining Edge Scoring with SimRank, and $\mathbf{A}^+$ is the adjacency matrix of the extended graph. The normalization operation ensures that the elements of $\widetilde{\mathbf{A}}$ are within the range $[0, 1]$.

Next, a weighted aggregation of the node features is performed, as GAT does. Specifically, the aggregation operation utilizes the attention weights $\widetilde{\mathbf{A}}^{(\ell)}$ at the-$l$ layer to compute a weighted sum of the node features, updating the feature representation for each node as follows

$$\mathbf{H}^{+(\ell+1)} = \sigma(\widetilde{\mathbf{A}}^{(\ell)}\mathbf{H}^{+(\ell)}\mathbf{W}^{(\ell)}), \tag{17}$$

$$\mathbf{H}^{+(0)} = \mathbf{X}^+, \tag{18}$$

where $\mathbf{H}^{+(\ell)}$ denotes the node feature matrix at the $\ell$-th layer, $\mathbf{X}^+$ is the node feature matrix of the extended graph, which incorporates both the node features of the original graph and the echo node features obtained via auto-encoding. $\widetilde{A}^{(\ell)}$ is the dynamic weight matrix, $\mathbf{W}^{(\ell)}$ is the feature transformation weights at the $\ell$-th layer, and $\sigma(\cdot)$ represents the nonlinear activation function.

## 4.3 Training

The training objective is to jointly minimize the reconstruction loss of the Variational Graph Auto-Encoder (VGAE) and the classification loss of the Graph Neural Network (GNN). In particular, the VGAE loss primarily consists of the reconstruction error and a regularization term, which quantifies the discrepancy between the generated graph structure and the original graph structure. The VGAE loss function $\mathcal{L}_{\text{vgae}}$ is expressed as:

$$\mathcal{L}_{\text{vgae}} = -\mathbb{E}_{q(Z|\mathbf{X},\mathbf{A})}[\log p(A|Z)] + \text{KL}(q(\mathbf{Z}|\mathbf{X},\mathbf{A}) \parallel p(Z)), \tag{19}$$

where $\mathbb{E}_{q(\mathbf{Z}|\mathbf{X},\mathbf{A})}[\log p(\mathbf{A}|\mathbf{Z})]$ represents the expected log-likelihood of reconstructing the adjacency matrix $A$ given the latent variable $\mathbf{Z}$. Here, $q(\mathbf{Z}|\mathbf{X},\mathbf{A})$ denotes the approximate posterior distribution, which is conditioned on the node features $X$ and the adjacency matrix $A$ of the original graph. The term $\text{KL}(q(\mathbf{Z}|\mathbf{X},\mathbf{A}) \parallel p(\mathbf{Z}))$ represents the Kullback-Leibler (KL) divergence between the approximate posterior distribution $q(\mathbf{Z}|\mathbf{X},\mathbf{A})$ and the prior distribution $p(\mathbf{Z})$. The KL divergence serves as a regularization term, encouraging $q(\mathbf{Z}|\mathbf{X},\mathbf{A})$ to remain close to the prior $p(\mathbf{Z})$.

The classification loss for our Echo-GAT is computed using cross-entropy loss $\mathcal{L}_{CE}$ for the node classification task. For each node $i$, the loss is calculated as

$$\mathcal{L}_{CE} = -\sum_{i \in V_0} y_i \log \hat{y}_i + (1 - y_i)\log(1 - \hat{y}_i), \tag{20}$$

where $V_0$ is the set of nodes in the original graph, $y_i$ is the true label of node $i$, and $\hat{y}_i$ is the predicted probability for node $i$.

Accordingly, the total loss function $\mathcal{L}_{\text{total}}$ is the weighted sum of the VGAE loss and the GNN classification loss

$$\mathcal{L}_{\text{total}} = a\mathcal{L}_{\text{vgae}} + b\mathcal{L}_{\text{CE}}. \tag{21}$$

## 4.4 Theoretical Analysis

Let $\mathcal{N}_{agu}(i) = \{v_1, v_2, \cdots, v_k\}$ be the echo neighbors of node $i$, then the augmented neighborhood is $\mathcal{N}'(i) = \mathcal{N}(i) \cup \{v_1, \ldots, v_k\} \cup \{i\}$. According to Eq. (8), the unnormalized attention score between $i$ and $v_m$ is

$\mathbf{e}_{iv_m} = \text{LeakyReLU}(\mathbf{a}^T[\mathbf{Wh}_i\|\mathbf{Wh}_{v_m}])$. Recall that the feature of echo node $m$ is copied from the central node initialized with VGAE, that is, $\mathbf{h}_{v_m}^{(l)} \approx \mathbf{h}_i^{(l)}$. Thus, the unnormalized attention score $e_{iv_m} \geq e_{ij}$ for heterophilic neighbor $j$.

Next, we analyze why the introduction of echo nodes decreases heterophilic influence on graph smoothing. First, we will see that adding terms with high $e_{iv_m}$ reduces the relative attention on low-similarity neighbors due to softmax normalization. Specifically, let $S = \sum_{k \in \mathcal{N}(i) \cup \{i\}} \exp(e_{ik})$ be the original denominator, and $S' = S + \sum_{m=1}^{k} \exp(e_{iv_m})$ the augmented one. For a heterophilic neighbor $j$ (corresponding to low $e_{ij}$), the original attention is $\alpha_{ij} = \frac{\exp(e_{ij})}{S}$, whereas the post-augmentation is

$$\alpha'_{ij} = \frac{\exp(e_{ij})}{S'} = \frac{\exp(e_{ij})}{S + \Delta}, \text{where} \quad \Delta = \sum_{m=1}^{k} \exp(e_{iv_m}). \tag{22}$$

Since $\Delta > 0$, $\alpha'_{ij} < \alpha_{ij}$ with the reduction factor $\frac{S}{S+\Delta} < 1$.

If echo nodes are highly similar, then $e_{iv_m} \approx e_{ii}$ and $\Delta$ can be written as $\Delta \approx k \exp(e_{ii})$. For large $k$ or high similarity, we have $\Delta \gg S - \exp(e_{ii})$, which makes $\alpha'_{ij} \ll \alpha_{ij}$ for heterophilic $j$. That is, *the augmentation of echo nodes dilutes the attention on heterophilic neighbors*. This effect further plays a role in reducing smoothing via aggregation bias.

In general, feature smoothing can be measured by the Dirichlet energy $\sum_i \|\mathbf{h}_i^{(l+1)} - \mathbf{h}_i^{(l)}\|^2$. Without loss of generality, we consider one term in the Dirichlet energy $\|\mathbf{h}_i^{(l+1)} - \mathbf{h}_i^{(l)}\|^2$. For convenience, we adopt a linear approximation of Eq. 16 and omit the activation $\sigma$ and weight matrix $\mathbf{W}$. So the aggregated feature is $\hat{\mathbf{h}}_i = \sum_j \alpha_{ij}\mathbf{h}_j^{(l)}$ before augmentation. Let $\mathcal{H}(i) \subset \mathcal{N}(i)$ be heterophilic subset. The heterophilic contribution is $\sum_{j \in \mathcal{H}(i)} \alpha_{ij}(\mathbf{h}_j^{(l)} - \mathbf{h}_i^{(l)})$, which biases $\hat{\mathbf{h}}_i$ away from $\mathbf{h}_i^{(l)}$.

When the central node $i$ is augmented with echo nodes, those echo nodes contribute $\sum_m \alpha'_{iv_m}(\mathbf{h}_{v_m}^{(l)} - \mathbf{h}_i^{(l)}) \approx 0$ (due to high similarity), while diluting $\alpha'_{ij}$ for all heterophilic neighbors $j \in \mathcal{H}(i)$. Such simplifications are commonly used in theoretical analyses of GCNs. Thus, the bias term shrinks to $\sum_{j \in \mathcal{H}(i)} \alpha'_{ij}(\mathbf{h}_j^{(l)} - \mathbf{h}_i^{(l)}) \approx \frac{S}{S+\Delta} \sum_{j \in \mathcal{H}(i)} \alpha_{ij}(\mathbf{h}_j^{(l)} - \mathbf{h}_i^{(l)})$, according to Eq. 22, theoretically bounding the Dirichlet energy increase per layer (reducing smoothing accordingly).

## 5 Experiments

This section presents experiments on nine widely used public datasets to evaluate the node classification accuracy and variance of the proposed Echo-GAT model, comparing it with benchmark models and those specifically designed for heterophilic graphs. Additionally, ablation studies, time complexity analyses, and parameter experiments are reported and analyzed. The datasets, comparison models, and implementation setup are described in detail in the appendix A. Codes are available at [URL will be provided upon acceptance].

### 5.1 Node Classification

**Heterophilic Graphs.** In Table 1, our method consistently delivers substantial performance improvements over vanilla GAT models across all six heterophilic benchmarks. In particular, Echo-GAT improves GAT by 61.92% on Cornell, 46.35% on Actor, and 23.83% on Tolokers, demonstrating its strong ability to handle heterophily-driven feature mixing. Even when compared with graph transformers such as GT, Echo-GAT still achieves notable gains—up to 5.97%, 2.60%, and 1.84% on Cornell, Texas, and Wisconsin, respectively. When compared with powerful heterophily-oriented methods, Echo-GAT achieves state-of-the-art (SOTA) performance on four out of six datasets and remains highly competitive on the remaining two.

**Homophilic Graphs.** As shown in Table 2, Echo-GAT achieves the best performance across all three homophilic datasets, surpassing both GAT and its variants. Although originally designed for heterophilic settings, our method generalizes effectively, achieving 83.09% on Cora and 71.53% on CiteSeer. These results

Table 1: Node classification accuracy (in percentage) on heterophily graph datasets. $h$ refers to the graph homophily level. The table reports mean accuracy ± standard deviation. Best results are red, second-best blue. ↑ / ↓ indicate improvement / degradation vs baseline models (%).

| | Datasets | Cornell | Texas | Wisconsin | Actor | Tolokers | Questions | Minesweeper | Avg.Rank |
| | $h$ | 0.13 | 0.11 | 0.20 | 0.22 | 0.19 | 0.11 | 0.25 | |
|---|---|---|---|---|---|---|---|---|---|
| **GAT variants** | GAT Velickovic et al. (2018) | 48.11 ± 5.1 | 59.46 ± 5.8 | 59.22 ± 5.9 | 28.38 ± 1.5 | 68.15 ± 0.4 | 63.75 ± 0.9 | 88.59 ± 0.5 | 16 |
| | GATv2 Brody et al. (2022) | 43.78 ± 6.1 | 61.08 ± 4.6 | 60.39 ± 3.9 | 26.86 ± 1.3 | 70.52 ± 0.3 | 65.20 ± 0.9 | 89.01 ± 0.4 | 15 |
| | GT Shi et al. (2021) | 73.51 ± 4.4 | 83.51 ± 5.3 | 85.10 ± 4.6 | 36.01 ± 0.9 | 78.16 ± 0.3 | 70.22 ± 1.1 | 91.85 ± 0.8 | 9 |
| **Heterophily-oriented** | **With Attention** | | | | | | | | |
| | WRGAT Suresh et al. (2021) | 76.49 ± 6.7 | 76.76 ± 4.0 | 79.61 ± 5.1 | 36.15 ± 1.0 | 83.00 ± 0.5 | 75.25 ± 1.1 | 89.75 ± 0.9 | 8 |
| | AERO-GNN Lee et al. (2023) | 76.76 ± 2.9 | 80.97 ± 5.6 | 79.33 ± 4.5 | 36.06 ± 1.0 | 83.08 ± 0.5 | 75.92 ± 1.2 | 89.88 ± 0.8 | 5 |
| | DIR-GAT Rossi et al. (2023) | 74.25 ± 3.6 | 81.25 ± 4.5 | 79.69 ± 3.5 | 36.28 ± 1.8 | 82.81 ± 0.8 | 74.12 ± 1.2 | 90.30 ± 1.0 | 7 |
| | **Without Attention** | | | | | | | | |
| | H2GCN Zhu et al. (2020) | 75.95 ± 7.3 | 75.68 ± 6.7 | 80.59 ± 2.9 | 36.23 ± 0.9 | 79.67 ± 0.5 | 75.76 ± 1.2 | 89.17 ± 0.3 | 11 |
| | CPGNN Zhu et al. (2021) | 70.27 ± 5.1 | 75.68 ± 5.1 | 76.47 ± 6.2 | 35.59 ± 0.9 | 77.59 ± 0.4 | 71.73 ± 1.0 | 87.43 ± 0.8 | 13 |
| | DGCN Tong et al. (2020b) | 68.32 ± 4.3 | 71.53 ± 7.2 | 65.52 ± 4.7 | 33.74 ± 0.3 | 76.32 ± 0.3 | 74.05 ± 0.9 | 88.57 ± 0.8 | 14 |
| | DiGCN Tong et al. (2020a) | 77.80 ± 4.9 | 79.50 ± 3.2 | 77.20 ± 2.2 | 32.82 ± 0.7 | 80.17 ± 0.7 | 74.36 ± 0.8 | 89.03 ± 0.7 | 10 |
| | GPR-GNN Chien et al. (2021) | 76.86 ± 7.1 | 81.51 ± 6.6 | 76.30 ± 3.9 | 35.58 ± 0.9 | 81.32 ± 0.4 | 75.09 ± 1.1 | 89.24 ± 0.6 | 6 |
| | GGCN Yan et al. (2022) | 71.35 ± 7.3 | 65.14 ± 8.3 | 74.12 ± 5.4 | 34.86 ± 0.9 | 77.10 ± 0.6 | 71.10 ± 1.3 | 87.69 ± 0.9 | 12 |
| | A2DUG Maekawa et al. (2023) | 74.59 ± 1.3 | 84.57 ± 1.1 | 78.35 ± 1.9 | 36.17 ± 0.9 | 83.53 ± 0.7 | 72.25 ± 0.9 | 89.97 ± 0.6 | 4 |
| | HeterGCL Wang et al. (2024a) | 74.69 ± 2.5 | 76.45 ± 4.0 | 78.76 ± 4.9 | 36.62 ± 1.4 | 83.77 ± 0.4 | 76.79 ± 1.4 | 90.50 ± 0.5 | 3 |
| | DCM Battiloro et al. (2024) | 77.80 ± 4.5 | 84.96 ± 5.6 | 85.05 ± 4.1 | 36.59 ± 2.2 | 84.21 ± 0.4 | 76.21 ± 1.6 | 90.63 ± 0.7 | 2 |
| **Ours** | **Echo-GAT** | 77.90 ± 2.5 | 85.68 ± 4.0 | 86.67 ± 2.9 | 36.89 ± 2.4 | 84.39 ± 0.3 | 77.55 ± 1.2 | 92.31 ± 0.6 | 1 |
| | vs GAT (%) | 61.92↑ | 44.10↑ | 46.35↑ | 29.99↑ | 23.83↑ | 21.65↑ | 4.20↑ | |
| | vs GATv2 (%) | 77.94↑ | 40.28↑ | 43.52↑ | 37.34↑ | 19.67↑ | 18.94↑ | 3.71↑ | |
| | vs GT (%) | 5.97↑ | 2.60↑ | 1.84↑ | 2.44↑ | 7.97↑ | 10.44↑ | 0.50↑ | |

Table 2: Node classification accuracy (in percentage) on homophilic graph datasets. $h$ refers to the homophily at graph level.

| Datasets | Cora | CiteSeer | PubMed | Computers | Photo |
| $h$ | 0.81 | 0.74 | 0.80 | 0.78 | 0.83 |
|---|---|---|---|---|---|
| GAT | 82.24 ± 0.7 | 70.19 ± 0.4 | 78.42 ± 0.3 | 81.90 ± 0.4 | 86.56 ± 1.2 |
| GATv2 | 82.62 ± 0.4 | 71.42 ± 0.7 | 78.34 ± 0.4 | 83.27 ± 0.3 | 88.15 ± 0.9 |
| GT | 82.09 ± 0.7 | 70.16 ± 0.8 | 79.04 ± 0.5 | 85.18 ± 0.5 | 92.74 ± 1.2 |
| H2GCN | 82.70 ± 0.9 | 43.70 ± 0.3 | 78.60 ± 0.6 | 83.36 ± 0.2 | 93.02 ± 0.8 |
| CPGNN | 79.40 ± 1.4 | 69.41 ± 0.5 | 77.40 ± 0.6 | 81.51 ± 0.4 | 85.87 ± 0.9 |
| **Echo-GAT** | 83.09 ± 0.3 | 71.53 ± 0.3 | 79.42 ± 0.5 | 88.45 ± 0.3 | 94.52 ± 0.8 |

suggest that the proposed attention debiasing mechanism also enhances feature aggregation in homophilic neighborhoods by mitigating noisy or uniform attention distributions, while preserving the inherent structural consistency.

## 5.2 Ablation Study

Table 3 reports the results of evaluating each core component of Echo-GAT by progressively removing Modules A (replacing the proposed Extended Attention with the vanilla graph attention), E (Edge Scoring), and S (SimRank). The complete model consistently achieves the highest accuracy across all datasets, underscoring the importance of the three modules working in synergy. Notably, removing Module A leads to a sharp decline in performance, highlighting the importance of echo nodes in redistributing attention within heterophilic graphs. Similarly, excluding Module E leads to substantial performance degradation on datasets, indicating the necessity of degree diversity-aware edge reweighting for distinguishing structurally correlated neighbors. Module S also proves essential by introducing higher-order structural relationships. These findings confirm that each module addresses a distinct yet complementary limitation of attention-based GNNs in heterophilic settings, and that their integration is key to achieving state-of-the-art performance.

We test whether the choice of the self-supervised encoder for echo node embedding will have effects on the prediction performance. We use three popular graph autoencoders, namely, VGAE, GraphMAE, and GraphMAE2 on four heterophilic graph datasets. From Table 4 we observe that across all datasets, our model based different graph encoders exhibits consistently high performance with only minor variations. This consistency suggests that our model is agnostic to the underlying embedding methods. Throughout the work, we use VGAE as the initial encoder.

Table 3: Ablation Study on Different Modules (A: Extended Attention, E: Edge Scoring, S: SimRank).

| A | E | S | Cornell | Texas | Wisconsin | Actor | Tolokers | Questions | Minesweeper |
|---|---|---|---|---|---|---|---|---|---|
| ✓ | ✓ | ✓ | 77.90 ± 2.5 | 85.68 ± 4.0 | 86.67 ± 2.9 | 36.89 ± 2.4 | 84.39 ± 0.3 | 77.55 ± 1.2 | 92.31 ± 0.6 |
| - | ✓ | ✓ | 52.16 ± 3.8 | 60.27 ± 4.8 | 58.63 ± 7.4 | 30.00 ± 7.1 | 69.81 ± 0.6 | 64.82 ± 1.5 | 82.05 ± 0.4 |
| ✓ | - | ✓ | 70.97 ± 5.8 | 77.03 ± 4.1 | 78.61 ± 3.7 | 34.39 ± 1.2 | 75.68 ± 0.2 | 71.09 ± 1.1 | 87.84 ± 0.7 |
| ✓ | ✓ | - | 75.95 ± 5.1 | 80.54 ± 5.1 | 79.63 ± 5.1 | 34.69 ± 1.3 | 77.51 ± 0.4 | 72.53 ± 1.2 | 89.20 ± 0.6 |

Table 4: Ablation Study on Different Autoencoders.

| Model | Cornell | Texas | Wisconsin | Actor |
|---|---|---|---|---|
| VGAE | 77.90 ± 2.5 | 85.68 ± 4.0 | 86.67 ± 2.9 | 36.89 ± 2.4 |
| GraphMAE | 77.89 ± 2.7 | 85.32 ± 4.3 | 86.49 ± 3.1 | 36.63 ± 0.9 |
| GraphMAE2 | 77.41 ± 3.5 | 85.66 ± 4.7 | 85.88 ± 3.4 | 36.65 ± 1.9 |

### 5.3 Effects of Hyperparameters

Fig. 4 illustrates the impact of varying the degree diversity threshold $\beta$ and the number of echo nodes $k$ added to each node on the accuracy. It is shown that the accuracy generally increases with $\beta$ up to approximately 80%, after which the improvement plateaus—likely due to the increasing density of the graphs. When the threshold is fixed, varying the value of $k$ does not significantly affect the model's performance , except for very small $k$ (i.e., $k = 1, 2$). These findings suggest that the effectiveness of echo node augmentation depends on the number of the nodes to be extended. Overall, the results highlight the importance of dataset-specific hyperparameter tuning to maximize the benefits of Echo-GAT.

### 5.4 Training Time

Fig. 5 shows the time cost in training phase for strong models with/without attention mechanism. It our method achieves an excellent balance between accuracy and efficiency. On the Cornell dataset, it reaches 77.90% accuracy with a training time of only $1.2 \times 10^{-2}$ seconds, significantly outperforming both GT and GPR-GNN. Similarly, on the Texas dataset, it achieves 85.68% accuracy with consistently low training cost. These results underscore the lightweight yet effective design of our approach, whose balanced performance and efficiency make it particularly suitable for scenarios with limited computational resources or requiring rapid iteration.

## 6 Conclusion

Heterophilic graphs pose a challenge to the current graph attention mechanism that quantifies the relevance between central node and its neighbors merely based on node features. Most of the existing solutions focus on determining the edge directionality, often overlooking the intrinsic correlation between node homophily and degree diversity. In this study, we investigate the performance bias of the vanilla graph attention networks towards inhomogeneous (thus heterophilic) nodes. We propose a novel attention extension paradigm to learn node feature representations more fairly, which introduce the echo nodes for homogeneous nodes to create attention diluting effect, thus enabling the nodes with similar feature/label to receive more attentions from the central nodes. We compare our method with the vanilla GAT and its variants, as well as heterophily-oriented GNNs with or without graph attention mechanism. We offer extensive experiments on nine benchmark datasets including both heterophilic and homophilious graphs. The results showcase the significant improvement over GAT variants and superiority compared to the state-of-the-art heterophily-oriented GNNs, highlighting the effectiveness of our method. Future research will seek to optimize the design of echo nodes to achieve higher computational efficiency.

## References

Sami Abu-El-Haija, Bryan Perozzi, Amol Kapoor, Nazanin Alipourfard, Kristina Lerman, Hrayr Harutyunyan, Greg Ver Steeg, and Aram Galstyan. Mixhop: Higher-order graph convolutional architectures via

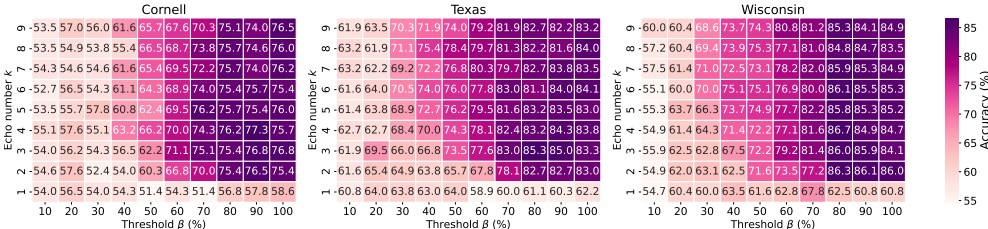

Figure 4: Impacts of hyperparameters on node classification.

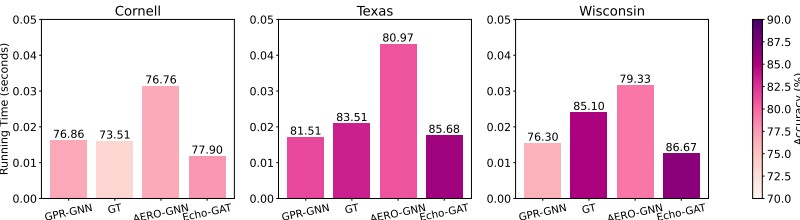

Figure 5: Training time vs. node classification performance for comparison methods. Lower histograms imply higher computational efficiency, while darker colors represent higher prediction effectiveness.

sparsified neighborhood mixing. In Kamalika Chaudhuri and Ruslan Salakhutdinov (eds.), *Proceedings of the 36th International Conference on Machine Learning, ICML 2019, 9-15 June 2019, Long Beach, California, USA*, volume 97 of *Proceedings of Machine Learning Research*, pp. 21–29. PMLR, 2019. URL `http://proceedings.mlr.press/v97/abu-el-haija19a.html`.

Guoguo Ai, Hui Yan, Huan Wang, and Xin Li. A2GCN: graph convolutional networks with adaptive frequency and arbitrary order. *Pattern Recognit.*, 156:110764, 2024. doi: 10.1016/J.PATCOG.2024.110764. URL `https://doi.org/10.1016/j.patcog.2024.110764`.

Claudio Battiloro, Indro Spinelli, Lev Telyatnikov, Michael M. Bronstein, Simone Scardapane, and Paolo Di Lorenzo. From latent graph to latent topology inference: Differentiable cell complex module. In *The Twelfth International Conference on Learning Representations, ICLR 2024, Vienna, Austria, May 7-11, 2024*, 2024. URL `https://openreview.net/forum?id=0JsRZEGZ7L`.

Deyu Bo, Xiao Wang, Chuan Shi, and Huawei Shen. Beyond low-frequency information in graph convolutional networks. *Proceedings of the AAAI Conference on Artificial Intelligence*, 35(5):3950–3957, May 2021. doi: 10.1609/aaai.v35i5.16514. URL `https://ojs.aaai.org/index.php/AAAI/article/view/16514`.

Aleksandar Bojchevski and Stephan Günnemann. Deep gaussian embedding of graphs: Unsupervised inductive learning via ranking. In *6th International Conference on Learning Representations, ICLR 2018, Vancouver, BC, Canada, April 30 - May 3, 2018, Conference Track Proceedings*, 2018. URL `https://openreview.net/forum?id=r1ZdKJ-0W`.

Shaked Brody, Uri Alon, and Eran Yahav. How attentive are graph attention networks? In *The Tenth International Conference on Learning Representations, ICLR 2022, Virtual Event, April 25-29, 2022*, 2022. URL `https://openreview.net/forum?id=F72ximsx7C1`.

Eli Chien, Jianhao Peng, Pan Li, and Olgica Milenkovic. Adaptive universal generalized pagerank graph neural network. In *9th International Conference on Learning Representations, ICLR 2021, Virtual Event, Austria, May 3-7, 2021*, 2021. URL `https://openreview.net/forum?id=n6jl7fLxrP`.

Haitz Sáez de Ocáriz Borde, Anees Kazi, Federico Barbero, and Pietro Liò. Latent graph inference using product manifolds. In *The Eleventh International Conference on Learning Representations, ICLR 2023, Kigali, Rwanda, May 1-5, 2023*, 2023. URL `https://openreview.net/forum?id=JLR_B7n_Wqr`.

Zhenyu Hou, Xiao Liu, Yukuo Cen, Yuxiao Dong, Hongxia Yang, Chunjie Wang, and Jie Tang. Graphmae: Self-supervised masked graph autoencoders. In Aidong Zhang and Huzefa Rangwala (eds.), *KDD '22: The 28th ACM SIGKDD Conference on Knowledge Discovery and Data Mining, Washington, DC, USA, August 14 - 18, 2022*, pp. 594–604. ACM, 2022. doi: 10.1145/3534678.3539321. URL https://doi.org/10.1145/3534678.3539321.

Zhenyu Hou, Yufei He, Yukuo Cen, Xiao Liu, Yuxiao Dong, Evgeny Kharlamov, and Jie Tang. Graphmae2: A decoding-enhanced masked self-supervised graph learner. In Ying Ding, Jie Tang, Juan F. Sequeda, Lora Aroyo, Carlos Castillo, and Geert-Jan Houben (eds.), *Proceedings of the ACM Web Conference 2023, WWW 2023, Austin, TX, USA, 30 April 2023 - 4 May 2023*, pp. 737–746. ACM, 2023. doi: 10.1145/3543507.3583379. URL https://doi.org/10.1145/3543507.3583379.

Weihua Hu, Matthias Fey, Marinka Zitnik, Yuxiao Dong, Hongyu Ren, Bowen Liu, Michele Catasta, and Jure Leskovec. Open graph benchmark: Datasets for machine learning on graphs. In Hugo Larochelle, Marc'Aurelio Ranzato, Raia Hadsell, Maria-Florina Balcan, and Hsuan-Tien Lin (eds.), *Advances in Neural Information Processing Systems 33: Annual Conference on Neural Information Processing Systems 2020, NeurIPS 2020, December 6-12, 2020, virtual*, 2020. URL https://proceedings.neurips.cc/paper/2020/hash/fb60d411a5c5b72b2e7d3527cfc84fd0-Abstract.html.

Jincheng Huang, Lun Du, Xu Chen, Qiang Fu, Shi Han, and Dongmei Zhang. Robust mid-pass filtering graph convolutional networks. In *Proceedings of the ACM Web Conference 2023*, pp. 328–338, 2023a.

Jincheng Huang, Ping Li, Rui Huang, Na Chen, and Acong Zhang. Revisiting the role of heterophily in graph representation learning: An edge classification perspective. *ACM Transactions on Knowledge Discovery from Data*, 18(1):1–17, 2023b.

Jincheng Huang, Jialie Shen, Xiaoshuang Shi, and Xiaofeng Zhu. On which nodes does GCN fail? enhancing GCN from the node perspective. In *Forty-first International Conference on Machine Learning, ICML 2024, Vienna, Austria, July 21-27, 2024*, 2024. URL https://openreview.net/forum?id=dcwUGaK9sQ.

Jincheng Huang, Yujie Mo, Xiaoshuang Shi, Lei Feng, and Xiaofeng Zhu. Enhancing the influence of labels on unlabeled nodes in graph convolutional networks. In *Forty-second International Conference on Machine Learning*, 2025.

Glen Jeh and Jennifer Widom. Simrank: a measure of structural-context similarity. In *Proceedings of the Eighth ACM SIGKDD International Conference on Knowledge Discovery and Data Mining, July 23-26, 2002, Edmonton, Alberta, Canada*, pp. 538–543. ACM, 2002. doi: 10.1145/775047.775126. URL https://doi.org/10.1145/775047.775126.

Mengying Jiang, Guizhong Liu, Yuanchao Su, and Xinliang Wu. Self-attention empowered graph convolutional network for structure learning and node embedding. *Pattern Recognit.*, 153:110537, 2024. doi: 10.1016/J.PATCOG.2024.110537. URL https://doi.org/10.1016/j.patcog.2024.110537.

Anees Kazi, Luca Cosmo, Seyed-Ahmad Ahmadi, Nassir Navab, and Michael M. Bronstein. Differentiable graph module (DGM) for graph convolutional networks. *IEEE Trans. Pattern Anal. Mach. Intell.*, 45 (2):1606–1617, 2023. doi: 10.1109/TPAMI.2022.3170249. URL https://doi.org/10.1109/TPAMI.2022.3170249.

Thomas N. Kipf and Max Welling. Variational graph auto-encoders. *CoRR*, abs/1611.07308, 2016. URL http://arxiv.org/abs/1611.07308.

Thomas N. Kipf and Max Welling. Semi-supervised classification with graph convolutional networks. In *5th International Conference on Learning Representations, ICLR 2017, Toulon, France, April 24-26, 2017, Conference Track Proceedings*, 2017. URL https://openreview.net/forum?id=SJU4ayYgl.

Soo Yong Lee, Fanchen Bu, Jaemin Yoo, and Kijung Shin. Towards deep attention in graph neural networks: Problems and remedies. In Andreas Krause, Emma Brunskill, Kyunghyun Cho, Barbara Engelhardt, Sivan Sabato, and Jonathan Scarlett (eds.), *International Conference on Machine Learning, ICML 2023, 23-29 July 2023, Honolulu, Hawaii, USA*, volume 202 of *Proceedings of Machine Learning Research*, pp. 18774–18795. PMLR, 2023. URL https://proceedings.mlr.press/v202/lee23b.html.

Xiang Li, Renyu Zhu, Yao Cheng, Caihua Shan, Siqiang Luo, Dongsheng Li, and Weining Qian. Finding global homophily in graph neural networks when meeting heterophily. In Kamalika Chaudhuri, Stefanie Jegelka, Le Song, Csaba Szepesvári, Gang Niu, and Sivan Sabato (eds.), *International Conference on Machine Learning, ICML 2022, 17-23 July 2022, Baltimore, Maryland, USA*, volume 162 of *Proceedings of Machine Learning Research*, pp. 13242–13256. PMLR, 2022. URL `https://proceedings.mlr.press/v162/li22ad.html`.

Derek Lim, Felix Hohne, Xiuyu Li, Sijia Linda Huang, Vaishnavi Gupta, Omkar Bhalerao, and Ser-Nam Lim. Large scale learning on non-homophilous graphs: New benchmarks and strong simple methods. In Marc'Aurelio Ranzato, Alina Beygelzimer, Yann N. Dauphin, Percy Liang, and Jennifer Wortman Vaughan (eds.), *Advances in Neural Information Processing Systems 34: Annual Conference on Neural Information Processing Systems 2021, NeurIPS 2021, December 6-14, 2021, virtual*, pp. 20887–20902, 2021. URL `https://proceedings.neurips.cc/paper/2021/hash/ae816a80e4c1c56caa2eb4e1819cbb2f-Abstract.html`.

Meng Liu, Hongyang Gao, and Shuiwang Ji. Towards deeper graph neural networks. In Rajesh Gupta, Yan Liu, Jiliang Tang, and B. Aditya Prakash (eds.), *KDD '20: The 26th ACM SIGKDD Conference on Knowledge Discovery and Data Mining, Virtual Event, CA, USA, August 23-27, 2020*, pp. 338–348. ACM, 2020. doi: 10.1145/3394486.3403076. URL `https://doi.org/10.1145/3394486.3403076`.

Yuankai Luo, Lei Shi, and Xiao-Ming Wu. Classic gnns are strong baselines: Reassessing gnns for node classification. In A. Globerson, L. Mackey, D. Belgrave, A. Fan, U. Paquet, J. Tomczak, and C. Zhang (eds.), *Advances in Neural Information Processing Systems*, volume 37, pp. 97650–97669. Curran Associates, Inc., 2024. doi: 10.52202/079017-3098. URL `https://proceedings.neurips.cc/paper_files/paper/2024/file/b10ed15ff1aa864f1be3a75f1ffc021b-Paper-Datasets_and_Benchmarks_Track.pdf`.

Yao Ma, Xiaorui Liu, Neil Shah, and Jiliang Tang. Is homophily a necessity for graph neural networks? In *The Tenth International Conference on Learning Representations, ICLR 2022, Virtual Event, April 25-29, 2022*, 2022. URL `https://openreview.net/forum?id=ucASPPD9GKN`.

Seiji Maekawa, Yuya Sasaki, and Makoto Onizuka. A simple and scalable graph neural network for large directed graphs. *arXiv preprint arXiv:2306.08274*, 2023.

Xuran Pan, Xiaoyan Han, Chaofei Wang, Zhuo Li, Shiji Song, Gao Huang, and Cheng Wu. A unified framework for convolution-based graph neural networks. *Pattern Recognit.*, 155:110597, 2024. doi: 10.1016/J.PATCOG.2024.110597. URL `https://doi.org/10.1016/j.patcog.2024.110597`.

Hongbin Pei, Bingzhe Wei, Kevin Chen-Chuan Chang, Yu Lei, and Bo Yang. Geom-gcn: Geometric graph convolutional networks. In *8th International Conference on Learning Representations, ICLR 2020, Addis Ababa, Ethiopia, April 26-30, 2020*, 2020. URL `https://openreview.net/forum?id=S1e2agrFvS`.

Oleg Platonov, Denis Kuznedelev, Michael Diskin, Artem Babenko, and Liudmila Prokhorenkova. A critical look at the evaluation of gnns under heterophily: Are we really making progress? In *The Eleventh International Conference on Learning Representations, ICLR 2023, Kigali, Rwanda, May 1-5, 2023*, 2023. URL `https://openreview.net/forum?id=tJbbQfw-5wv`.

Emanuele Rossi, Bertrand Charpentier, Francesco Di Giovanni, Fabrizio Frasca, Stephan Günnemann, and Michael M. Bronstein. Edge directionality improves learning on heterophilic graphs. In Soledad Villar and Benjamin Chamberlain (eds.), *Learning on Graphs Conference, 27-30 November 2023, Virtual Event*, volume 231 of *Proceedings of Machine Learning Research*, pp. 25. PMLR, 2023. URL `https://proceedings.mlr.press/v231/rossi24a.html`.

Benedek Rozemberczki, Paul Scherer, Yixuan He, George Panagopoulos, Alexander Riedel, Maria Sinziana Astefanoaei, Oliver Kiss, Ferenc Béres, Guzmán López, Nicolas Collignon, and Rik Sarkar. Pytorch geometric temporal: Spatiotemporal signal processing with neural machine learning models. In Gianluca Demartini, Guido Zuccon, J. Shane Culpepper, Zi Huang, and Hanghang Tong (eds.), *CIKM '21: The 30th ACM International Conference on Information and Knowledge Management, Virtual Event, Queensland, Australia, November 1 - 5, 2021*, pp. 4564–4573. ACM, 2021. doi: 10.1145/3459637.3482014. URL `https://doi.org/10.1145/3459637.3482014`.

Yunsheng Shi, Zhengjie Huang, Shikun Feng, Hui Zhong, Wenjing Wang, and Yu Sun. Masked label prediction: Unified message passing model for semi-supervised classification. In Zhi-Hua Zhou (ed.), *Proceedings of the Thirtieth International Joint Conference on Artificial Intelligence, IJCAI 2021, Virtual Event / Montreal, Canada, 19-27 August 2021*, pp. 1548–1554. ijcai.org, 2021. doi: 10.24963/IJCAI.2021/214. URL https://doi.org/10.24963/ijcai.2021/214.

Yunchong Song, Chenghu Zhou, Xinbing Wang, and Zhouhan Lin. Ordered GNN: ordering message passing to deal with heterophily and over-smoothing. In *The Eleventh International Conference on Learning Representations, ICLR 2023, Kigali, Rwanda, May 1-5, 2023*, 2023. URL https://openreview.net/forum?id=wKPmPBHSnT6.

Yukuan Sun, Yutai Duan, Haoran Ma, Yuelong Li, and Jianming Wang. High-frequency and low-frequency dual-channel graph attention network. *Pattern Recognit.*, 156:110795, 2024. doi: 10.1016/J.PATCOG.2024.110795. URL https://doi.org/10.1016/j.patcog.2024.110795.

Susheel Suresh, Vinith Budde, Jennifer Neville, Pan Li, and Jianzhu Ma. Breaking the limit of graph neural networks by improving the assortativity of graphs with local mixing patterns. In Feida Zhu, Beng Chin Ooi, and Chunyan Miao (eds.), *KDD '21: The 27th ACM SIGKDD Conference on Knowledge Discovery and Data Mining, Virtual Event, Singapore, August 14-18, 2021*, pp. 1541–1551. ACM, 2021. doi: 10.1145/3447548.3467373. URL https://doi.org/10.1145/3447548.3467373.

Zekun Tong, Yuxuan Liang, Changsheng Sun, Xinke Li, David S. Rosenblum, and Andrew Lim. Digraph inception convolutional networks. In Hugo Larochelle, Marc'Aurelio Ranzato, Raia Hadsell, Maria-Florina Balcan, and Hsuan-Tien Lin (eds.), *Advances in Neural Information Processing Systems 33: Annual Conference on Neural Information Processing Systems 2020, NeurIPS 2020, December 6-12, 2020, virtual*, 2020a. URL https://proceedings.neurips.cc/paper/2020/hash/cffb6e2288a630c2a787a64ccc67097c-Abstract.html.

Zekun Tong, Yuxuan Liang, Changsheng Sun, David S. Rosenblum, and Andrew Lim. Directed graph convolutional network. *CoRR*, abs/2004.13970, 2020b. URL https://arxiv.org/abs/2004.13970.

Petar Velickovic, Guillem Cucurull, Arantxa Casanova, Adriana Romero, Pietro Liò, and Yoshua Bengio. Graph attention networks. In *6th International Conference on Learning Representations, ICLR 2018, Vancouver, BC, Canada, April 30 - May 3, 2018, Conference Track Proceedings*, 2018. URL https://openreview.net/forum?id=rJXMpikCZ.

Chenhao Wang, Yong Liu, Yan Yang, and Wei Li. Hetergcl: Graph contrastive learning framework on heterophilic graph. In *Proceedings of the Thirty-Third International Joint Conference on Artificial Intelligence, IJCAI 2024, Jeju, South Korea, August 3-9, 2024*, pp. 2397–2405. ijcai.org, 2024a. URL https://www.ijcai.org/proceedings/2024/265.

Junfu Wang, Yuanfang Guo, Liang Yang, and Yunhong Wang. Heterophily-aware graph attention network. *Pattern Recognit.*, 156:110738, 2024b. doi: 10.1016/J.PATCOG.2024.110738. URL https://doi.org/10.1016/j.patcog.2024.110738.

Tao Wang, Di Jin, Rui Wang, Dongxiao He, and Yuxiao Huang. Powerful graph convolutional networks with adaptive propagation mechanism for homophily and heterophily. In *Thirty-Sixth AAAI Conference on Artificial Intelligence, AAAI 2022, Thirty-Fourth Conference on Innovative Applications of Artificial Intelligence, IAAI 2022, The Twelveth Symposium on Educational Advances in Artificial Intelligence, EAAI 2022 Virtual Event, February 22 - March 1, 2022*, pp. 4210–4218. AAAI Press, 2022. doi: 10.1609/AAAI.V36I4.20340. URL https://doi.org/10.1609/aaai.v36i4.20340.

Felix Wu, Amauri H. Souza Jr., Tianyi Zhang, Christopher Fifty, Tao Yu, and Kilian Q. Weinberger. Simplifying graph convolutional networks. In Kamalika Chaudhuri and Ruslan Salakhutdinov (eds.), *Proceedings of the 36th International Conference on Machine Learning, ICML 2019, 9-15 June 2019, Long Beach, California, USA*, volume 97 of *Proceedings of Machine Learning Research*, pp. 6861–6871. PMLR, 2019. URL http://proceedings.mlr.press/v97/wu19e.html.

Keyulu Xu, Chengtao Li, Yonglong Tian, Tomohiro Sonobe, Ken-ichi Kawarabayashi, and Stefanie Jegelka. Representation learning on graphs with jumping knowledge networks. In Jennifer G. Dy and Andreas Krause (eds.), *Proceedings of the 35th International Conference on Machine Learning, ICML 2018, Stockholmsmässan, Stockholm, Sweden, July 10-15, 2018*, volume 80 of *Proceedings of Machine Learning Research*, pp. 5449–5458. PMLR, 2018. URL `http://proceedings.mlr.press/v80/xu18c.html`.

Lixiang Xu, Jiawang Peng, Xiaoyi Jiang, Enhong Chen, and Bin Luo. Graph neural network based on graph kernel: A survey. *Pattern Recognit.*, 161:111307, 2025. doi: 10.1016/J.PATCOG.2024.111307. URL `https://doi.org/10.1016/j.patcog.2024.111307`.

Yujun Yan, Milad Hashemi, Kevin Swersky, Yaoqing Yang, and Danai Koutra. Two sides of the same coin: Heterophily and oversmoothing in graph convolutional neural networks. In Xingquan Zhu, Sanjay Ranka, My T. Thai, Takashi Washio, and Xindong Wu (eds.), *IEEE International Conference on Data Mining, ICDM 2022, Orlando, FL, USA, November 28 - Dec. 1, 2022*, pp. 1287–1292. IEEE, 2022. doi: 10.1109/ICDM54844.2022.00169. URL `https://doi.org/10.1109/ICDM54844.2022.00169`.

Zhilin Yang, William W. Cohen, and Ruslan Salakhutdinov. Revisiting semi-supervised learning with graph embeddings. In Maria-Florina Balcan and Kilian Q. Weinberger (eds.), *Proceedings of the 33nd International Conference on Machine Learning, ICML 2016, New York City, NY, USA, June 19-24, 2016*, volume 48 of *JMLR Workshop and Conference Proceedings*, pp. 40–48. JMLR.org, 2016. URL `http://proceedings.mlr.press/v48/yanga16.html`.

Jiong Zhu, Yujun Yan, Lingxiao Zhao, Mark Heimann, Leman Akoglu, and Danai Koutra. Beyond homophily in graph neural networks: Current limitations and effective designs. In Hugo Larochelle, Marc'Aurelio Ranzato, Raia Hadsell, Maria-Florina Balcan, and Hsuan-Tien Lin (eds.), *Advances in Neural Information Processing Systems 33: Annual Conference on Neural Information Processing Systems 2020, NeurIPS 2020, December 6-12, 2020, virtual*, 2020. URL `https://proceedings.neurips.cc/paper/2020/hash/58ae23d878a47004366189884c2f8440-Abstract.html`.

Jiong Zhu, Ryan A. Rossi, Anup Rao, Tung Mai, Nedim Lipka, Nesreen K. Ahmed, and Danai Koutra. Graph neural networks with heterophily. In *Thirty-Fifth AAAI Conference on Artificial Intelligence, AAAI 2021, Thirty-Third Conference on Innovative Applications of Artificial Intelligence, IAAI 2021, The Eleventh Symposium on Educational Advances in Artificial Intelligence, EAAI 2021, Virtual Event, February 2-9, 2021*, pp. 11168–11176. AAAI Press, 2021. doi: 10.1609/AAAI.V35I12.17332. URL `https://doi.org/10.1609/aaai.v35i12.17332`.

# A Appendix

## A.1 Datasets.

We conduct experiments on nine datasets from PyTorch Geometric (Rozemberczki et al., 2021), including three homophilic and six heterophilic datasets, as summarized in Appendix Table 5. The Homophily ratio $h$ quantifies the proportion of edges connecting nodes with the same label—values close to 1 indicate strong homophily, while values near 0 reflect heterophily. The Wisconsin, Cornell, and Texas datasets are subsets of WebKB (Pei et al., 2020), collected by Carnegie Mellon University, where nodes represent web pages and edges correspond to hyperlinks. Tolokers (Platonov et al., 2023) is a user–task interaction network derived from a crowdsourcing platform, where nodes represent users and edges denote that two users have worked on the same task. Questions (Platonov et al., 2023) is a question–answer interaction network.The Actor dataset is a subgraph of the movie-director-actor-writer network, where nodes represent actors and edges denote co-occurrence on the same Wikipedia page. Minesweeper (Platonov et al., 2023) is a synthetic graph dataset derived from the classic Minesweeper game, where nodes correspond to grid cells and edges connect spatially adjacent cells, and node labels indicate the number of mines in neighboring cells. Cora (Bojchevski & Günnemann, 2018), CiteSeer (Yang et al., 2016), and PubMed (Hu et al., 2020) are citation networks, where nodes correspond to research papers and edges represent citation relationships. Finally, Computers and Photo (Luo et al., 2024) are Amazon co-purchase networks, where nodes represent products and edges denote that two products are frequently co-purchased, and node labels correspond to product categories.

Table 5: Datasets

| Datasets | Nodes | Edges | Features | Train/Val/Test | $h$ |
|---|---|---|---|---|---|
| Cornell | 183 | 298 | 1703 | 48%/32%/20% | 0.13 |
| Texas | 183 | 325 | 1703 | 48%/32%/20% | 0.11 |
| Wisconsin | 251 | 515 | 1703 | 48%/32%/20% | 0.20 |
| Actor | 7600 | 30019 | 932 | 48%/32%/20% | 0.22 |
| Tolokers | 11758 | 519000 | 10 | 50%/25%/25% | 0.19 |
| Questions | 48921 | 153540 | 301 | 50%/25%/25% | 0.11 |
| Minesweeper | 10000 | 39402 | 7 | 50%/25%/25% | 0.25 |
| Cora | 2708 | 10556 | 1433 | 9%/30%/61% | 0.81 |
| CiteSeer | 3327 | 9104 | 3703 | 7%/31%/62% | 0.74 |
| PubMed | 19717 | 88648 | 500 | 4%/32%/64% | 0.80 |
| Computers | 13752 | 505474 | 767 | 60%/20%/20% | 0.78 |
| Photo | 7650 | 245812 | 745 | 60%/20%/20% | 0.83 |

## A.2 Comparison methods.

We compare our Echo-GAT model with several graph attention based network models (including strong graph transformer) and twelve state-of-the-art heterophily-oriented methods, three of which incorporate attention mechanisms. These methods are as follows: GAT (Velickovic et al., 2018), GATv2 (Brody et al., 2022), and GT (Shi et al., 2021) utilize various attention mechanisms, including Transformer self-attention, to dynamically weigh neighbor contributions. Approaches for directed graphs, such as DGCN (Tong et al., 2020b), DiGCN (Tong et al., 2020a), and GPR-GNN (Chien et al., 2021), incorporate proximity measures and adaptive PageRank weights to capture multi-scale features and prevent over-smoothing. Other methods, including WRGAT (Suresh et al., 2021), AERO-GNN (Lee et al., 2023), DIR-GAT (Rossi et al., 2023), H2GCN (Zhu et al., 2020), CPGNN (Zhu et al., 2021), GGCN (Yan et al., 2022), A2DUG (Maekawa et al., 2023), and HeterGCL (Wang et al., 2024a), address challenges such as heterophily and edge directionality by leveraging multi-relational graphs, adaptive attention, separate aggregation of incoming and outgoing edges, and contrastive learning. Finally, DCM (Battiloro et al., 2024) introduces a Differentiable Cell Complex Module to infer higher-order topologies, advancing graph representation learning.

## A.3 Implementation settings.

For all comparison methods, we retain the original parameter settings as reported in their respective studies. Each dataset is trained, validated, and tested using the ten splits provided by PyTorch Geometric. All experiments are conducted using PyTorch version 2.3.0, the Adam optimizer, an NVIDIA RTX 4090 GPU, and CUDA version 12.1. In our proposed model, Echo-GAT, the predefined threshold $\beta$ represents the degree diversity threshold and is searched within the range $[0.1, 0.9]$. The number of added echo nodes $k$ is selected from the set $\{1, 2, 3, 4, 5, 6, 7, 8, 9, 10\}$. Echo-GAT uses the Adam optimizer, with the learning rate searched in the range $[0.0001, 0.1]$, the number of hidden units chosen from $\{64, 128, 256, 512\}$, and the weight decay fixed at 0.0005. An early stopping strategy is applied based on both validation and test performance. Echo-GAT is implemented using the PyTorch framework.

