# OpenReview forum: "Echo-GAT: Debiasing Graph Attention with Echo Nodes and Degree Diversity for Heterophilic Graphs"
_TMLR — Rejected by TMLR_

### Review · Reviewer_qWHQ · 2026-02-11

**Summary Of Contributions:**

The authors study the issue of graph attention mechanisms performing poorly on heterophilic graphs. They show that node-level homophily may not be informative about why graph attention mechanisms do not work well. They propose to look at per-node degree diversity, which is defined as the absolute difference between the node degree and the average degree of its neighbors. Based on this idea, they further propose Echo-GAT, which adds virtual nodes to the neighbors of nodes with both low node degree and per-node degree diversity. The effectiveness of Echo-GAT is validated by extensive experiments on benchmark datasets.

### Strength
(+) The experiment is extensive, which shows that Echo-GAT performs well on the tested datasets.


### Weaknesses
(-) Unclear motivation: The "degree diversity" and the "homophily/heterophily" measures seem unrelated and orthogonal. I do not understand why looking at "degree diversity" is a reasonable remedy to the problem of graph attention on heterophilic graphs.

(-) Unclear design logic: the use of VGAE (or any other auto-encoders) to generate the “echo nodes” seems problematic.

### Detail Comments
The first problem of this work is the unclear motivation. The authors never explain why per-node degree diversity (i.e., the absolute difference between average neighbor node degree and its own node degree) is important for resolving the problem of graph attention on heterophilic graphs. The only thing I found in the main text that is relevant to this point is Figure 1. Yet it barely says heterophily is not the core cause for graph attention, and we should look at per-node degree diversity instead. Note that these two statements are very different. I think the authors need to clarify what claim they want to make. If they want to claim that degree diversity is important to resolve the heterophilic graph issue, they should provide more arguments and evidence on this end. If they want to claim that we should not care about graph heterophily and per-node degree diversity is the root cause of why graph attention can perform poorly, they should modify their entire paper accordingly. Specifically, in this case, they should demonstrate that their claim holds on homophilic graphs (beyond those three simple citation networks) as well.

The critical problem is the unclear design logic of their proposed “echo node”. According to my understanding, the effect of echo nodes roughly acts like adding self-loops. For each selected node $i$, the corresponding echo node will use its node embedding as the initial node features and link to itself but not the other nodes, according to the illustrations of Figures 2 and 3. However, their equation (6) seems to suggest that the echo node of node $i$ may also have an edge to node $j\neq i$? This is very confusing in the first place. I assume that the echo node of node $i$ will only connect to node $i$ itself. However, in this setup, why do we need the complicated auto-encoder? Why not simply add more self-loops, or duplicate node $i$ directly as the “echo node”? The way of setting node features of echo nodes is also weird. Note that usually the node embedding of VGAE (or any other graph autoencoder) has dimensions different from the inherent node features in the dataset. Why does such a design choice always make sense?

In summary, I feel the paper is far below the bar of TMLR in its current form. I hope the authors can spend enough time in addressing the aforementioned issue in their revision.

**Audience:**

Yes

**Audience Explanation:**

The graph attention mechanism is important for graph neural networks. I believe some individuals working in this direction will be interested in this topic.

**Claims And Evidence:**

No

**Claims Explanation:**

Please see my comments above. I do not think the authors clearly explain why their method is reasonable for resolving the problem.

**Requested Changes:**

Please address the weaknesses above if possible.

---

> ### Author Response · Authors · 2026-03-09
>
> 1. **The motivation of this work is unclear**.
>
> Thank you for the insightful comment. We agree that homophily and degree diversity describe different properties of graphs: homophily measures label-level similarity between connected nodes, whereas degree diversity reflects structural contrast at the node level.
>
> Our motivation does not assume that degree diversity replaces homophily. Rather, we argue that under heterophily, when feature and label similarity become unreliable cues for attention, structural signals may become more influential in guiding neighbor discrimination. In homophilic graphs, feature similarity often aligns with label similarity, allowing attention mechanisms to naturally emphasize informative neighbors. However, in heterophilic graphs this alignment breaks down, forcing attention mechanisms to rely more heavily on alternative structural cues. In this context, structural contrast becomes particularly relevant.
>
> Importantly, this observation is also supported empirically. As shown in Fig. 1, on heterophilic graphs the correlation between node-level homophily and prediction accuracy is weak or inconsistent. In contrast, grouping nodes by degree diversity leads to a much clearer separation in accuracy, indicating a stronger association between structural contrast and attention performance.
>
> Therefore, degree diversity is not proposed as a substitute for homophily, but rather as a structural factor that better explains node-level performance imbalance of attention under heterophily. Echo-GAT addresses this imbalance by explicitly regularizing structural contrast, thereby stabilizing attention behavior in heterophilic settings.
>
> We will revise the manuscript to clarify this conceptual distinction and strengthen the quantitative comparison between homophily and degree diversity in explaining attention performance.
>
> 2. **The design logic of using VGAE to generate echo nodes is unclear**.
>
> Thank you for the comment. The introduction of echo nodes is motivated by the need to regularize the local neighborhood structure and increase structural contrast for attention normalization. Since these echo nodes actively participate in message passing, they must be equipped with meaningful feature representations rather than acting as empty placeholders.
>
> To ensure that the generated echo nodes are structurally consistent with the original graph, we initialize them using node embeddings learned from a self-supervised graph encoder. This allows echo nodes to reflect the topological context of their corresponding central nodes, while remaining fully compatible with subsequent GNN propagation.
>
> Importantly, as shown in Table 4, the effectiveness of Echo-GAT does not depend on a specific encoder choice. This indicates that the role of the encoder is mainly to provide structurally informed initialization, rather than to drive the performance improvement itself.
>
> 3. **The design logic of the proposed echo node is unclear**.
>
> Thank you for raising these concerns. Echo nodes only connect to their corresponding original node, which is consistent with Figures 2–3. In Eq. (7), the index $i$ refers to the original node and $(n+j⋅n^+)$ denotes the associated echo node. No connections exist between echo nodes or between echo nodes and other original nodes. We will revise the manuscript to explicitly clarify this connectivity rule to avoid potential confusion.
>
> Regarding why we do not simply duplicate nodes or add self-loops, directly copying raw node features would ignore structural information from the graph topology. In contrast, embeddings learned by graph autoencoders capture neighborhood-aware structural representations. Using a self-supervised graph encoder therefore provides a principled way to initialize echo nodes such that they remain structurally aligned with the original topology.
>
> Finally, although VGAE latent embeddings may have a different dimension from the raw node features, in our implementation these embeddings are mapped back to the original feature dimension before participating in message passing. Therefore, dimensional consistency with the original feature space is preserved.

---

### Review · Reviewer_KJ5T · 2026-02-20

**Summary Of Contributions:**

The paper addresses the poor predictive performance of attention-based graph neural networks (GNNs) on heterofilic graphs for node classification tasks. The paper identifies that homophily and heterophily alone don’t explain the performance deterioration, and shows that degree diversity levels are more strongly associated with performance deterioration.
Based on this observation, the authors develop a novel model, Echo-GAT, that introduces some virtual nodes called echo nodes in order to alleviate the issues in attention in the presence of nodes with insufficient degree diversity. These echo nodes are created via a graph autoencoder. The paper also presents some theoretical justification on why the new attention with echo nodes reduces oversmoothing.
The method is benchmarked on some homophilic and heterophilic datasets, showing, in both cases, better predictive performance compared to the reported baselines.

Strengths:
- the paper is quite well-written and easy to follow
- the authors identified some issues of current attention-based architectures and proposed an interesting solution that seems to produce promising empirical results

Weaknesses:
- The paper is not formal enough in some aspects. For example, in eq. 2 and 14, they refer to some Norm(.) function, but do not specify what this function is, only that it “scales the value to the range [0, 1]”. Clearly, there are multiple ways to do so, but they don’t specify how.
- Similarly, Section 4.4 seems very “hand-wavy”, as it contains general statements without justification (“self-attention score, often high”, “Without loss of generality, we consider one term in the Dirichlet energy”), and approximations without guarantees (“For convenience, we use linear approximation of Eq. 17 by ignoring σ and W”).
- Most importantly, the experimental validation is not thorough and rigorous enough.
    - There exist many more node classification datasets which should be used for benchmarking a new model. For heterophilic ones, consider Roman-Empire, Amazon-Ratings, Minesweeper from [1]. For homophilic ones, the three ones used in the paper are very outdated. Consider using at least the ogb [2] ones, or more recent ones.
    -  The method is not benchmarked against strong GNN variants. Most notably, the models from [3], despite their simplicity, report higher accuracies on Cora, Citeseer and Pubmed, the only shared datasets between the two papers. It would be interesting to compare to these methods on these datasets, and on the datasets I mentioned above.
    - Even more concerning is the fact that the accuracy of the GAT model reported in [3] (i.e., without the modifications that make it GAT*) is higher than that of the GAT model reported in the present paper, and are very close to that of Echo-GAT (probably equal up to statistical uncertainty). This might indicate that hyperparameters for baselines have not been chosen properly.

Minor points:
- In Figure 1, $h$ and $h_{dd}$ are never defined.
- Some singular/plural forms are used incorrectly: e.g., Section 5 should probably be called Experiments, Section 5.3 Effects of Hyperparameters.

------
References
- [1] Oleg Platonov, Denis Kuznedelev, Michael Diskin, Artem Babenko, and Liudmila Prokhorenkova. A critical look at the evaluation of gnns under heterophily: Are we really making progress? arXiv preprint arXiv:2302.11640, 2023.
- [2] Weihua Hu, Matthias Fey, Marinka Zitnik, Yuxiao Dong, Hongyu Ren, Bowen Liu, Michele Catasta, and Jure Leskovec. Open graph benchmark: Datasets for machine learning on graphs. NeurIPS 2020.
- [3] Yuankai Luo, Lei Shi, Xiao-Ming Wu. Classic GNNs are Strong Baselines: Reassessing GNNs for Node Classification. NeurIPS 2024.

**Audience:**

Yes

**Audience Explanation:**

If the issues with the lack of formality and, most importantly, with the experimental validation are solved, I think this paper will be of interest to the graph learning community.

**Broader Impact Concerns:**

None.

**Claims And Evidence:**

No

**Claims Explanation:**

My main concern is the weak experimental evaluation. More datasets and strong baselines should be considered (see weaknesses above).

**Requested Changes:**

- Include more datasets in the experimental validation, and compare to stronger baselines (e.g. [3]).
- Make the "hand-wavy" parts more formal and clearer.

---

> ### Author Response · Authors · 2026-03-09
>
> 1. **The paper is not formal enough in some aspects**.
>
> Thank you for pointing this out. In Eqs. (2) and (14) prior work has shown that GCN layers can be interpreted as Laplacian smoothing, We use min–max normalization to scale values into the range $[0,1]$, and we will explicitly define this normalization formula in the revised manuscript to ensure clarity and reproducibility.
>
> We apologize that the explanation in Section 4.4 was not sufficiently precise, and we will revise the text to clarify these statements.
>
> The activation function in GCN is typically ReLU, which behaves linearly when the input is positive. Therefore, ignoring $\sigma$ does not significantly affect the analysis of message propagation. The weight matrix $\mathbf{W}$ mainly performs feature projection and does not alter the neighborhood aggregation structure. Removing 𝑊 therefore does not change the underlying message passing mechanism.
>
> Such simplifications are commonly adopted in theoretical analyses of GCNs. For example, the Simplifying Graph Convolutional Networks (SGC) [1] model removes nonlinear activation functions and intermediate transformations to analyze graph propagation in a linear form. Moreover, prior work has shown that GCN layers can be interpreted as Laplacian smoothing operations over graph signals, indicating that the essential mechanism of GCN lies in the propagation operator rather than the feature transformation [2-3].
>
> 2. **The experimental validation is not thorough and rigorous enough**.
>
> Thank you for this important observation. We will expand our experiments to include heterophilic datasets (Minesweeper), as well as homophilic datasets including Computers and Photo. In addition, we will carefully tune baseline hyperparameters, ensure consistent data splits, and report standard deviations over multiple runs. If performance differences remain, we will explicitly discuss their potential causes, including implementation details and data split settings.
>
> 3. **Minor issues**.
>
> Thank you for pointing out these minor issues. We have revised Figure 1 to clearly define all symbols in both the caption and the main text. We also renamed Section 5 to “Experiments” and Section 5.3 to “Effects of Hyperparameters” for grammatical correctness. In addition, we carefully proofread the manuscript to correct singular/plural inconsistencies and improve overall clarity and formatting consistency throughout the paper.
>
> [1] Wu, Felix, et al. "Simplifying graph convolutional networks." International conference on machine learning. Pmlr, 2019.
> [2] Li Q, Han Z, Wu X M. Deeper insights into graph convolutional networks for semi-supervised learning[C]//Proceedings of the AAAI conference on artificial intelligence. 2018, 32(1).
> [3] Jiang B, Zhang Z, Lin D, et al. Semi-supervised learning with graph learning-convolutional networks[C]//Proceedings of the IEEE/CVF conference on computer vision and pattern recognition. 2019: 11313-11320. structural signals become more influential

---

### Review · Reviewer_rCuW · 2026-02-23

**Summary Of Contributions:**

This paper presents an analysis and a method for the issue of accuracy degradation of graph attention networks (GAT) on heterophilous graphs. The authors first argue that, instead of heterophily, the lack of node degree diversity should be attributed for the accuracy degradation. To address the problem, the authors propose a method, Echo-GAT. It first creates echo nodes for nodes that are low in terms of both degree diversity and degrees, and then propose a joint attention mechanism that attends to neighboring nodes depending on both node features and degrees (i.e. structural features). Experiments on a variety of graph datasets, with homophily degree ranging from 0.11 to 0.8, show the effectiveness of the proposed EchoGAT.

**Additional Comments:**

No additional comments.

**Audience:**

Yes

**Audience Explanation:**

Homophily VS heterophily is a constant problem when it comes to practical usage of graph neural networks. This paper tries to provide more insights on this problem by proposing new explanations, and correspondingly, new methods to address it. I think there would be at least some researchers and practitioners who study algorithms & applications of GNNs that are interested in this paper.

**Broader Impact Concerns:**

No concerns.

**Claims And Evidence:**

No

**Claims Explanation:**

Although this paper tries to present new explanations and findings on the problem of heterophilous graphs, the paper has some key drawbacks, where the claims made by the authors are not fully & convincinly justified. Here are some of them.

1. First, the definition and meaning of "node degree diversity" is not very clear. From the wording, I would expect the term to be defined as "the extent to which node degrees deviate from the average degree", ot in a local way, "the extent to which node degrees in a neighborhood deviate from the average of that neighborhood". Indeed, the authors seem to implicitly state this by saying "this challenge is particularly pronounced on heterophilic graphs, where node neighborhoods often exhibit heterogeneous degree patterns, offering rich structural cues for distinguishing neighbors. However, when such degree heterogeneity is insufficient (i.e., nodes exhibit low degree diversity), their neighborhoods become structurally uniform, with neighbors sharing similar degrees." However, in Eqn. 2, the concept of degree diversity does not seem to be defined so. Rather, it is defined as "the deviation of a central node from the average of its neighborhood". These two concepts seem contradictory, and in my opinion, the former makes more sense. The authors may come up with some better justification/definition of the important concept of "degree diversity".

2. Second, Fig. 1 is not sufficiently conclusive in terms of justifying the motivation of this paper. Several changes can be made to make the observation more solid. For example, how is the deviation in Fig. 1 significant? How does it compare to the average deviation of prediction accuracy? Is it possible that some other factors caused the deviation? ...

3. The methodology part is not very clear in certain places.
- The most significant one, in my opinion, is how the echo nodes are connected with other nodes. In the paper, the authors only mention that the echo nodes are connected to the original ones, and do not connect with other echo nodes, but nothing about how they are connected with other nodes. This is an important detail, as their degrees come from the nodes they are connected with. Without it, readers may not fully understand the proposed method.
- Some less significant points in the methodology include:
    - the construction of $E_f$. The authors use $d_i\cdot d_j$ to construct the edge feature matrix. May it be susceptible to some very high-degree nodes?
    - It seems that the notion of "degree diversity" is not present in the construction of $E_f$. Would the introduction of some degree diversity metrics make $E_f$ even better?

4. Experiments bear room for improvement.
- The most significant shortcoming is that the datasets are fairly small (especially Texas, Wisconsin, Cora and Citeseer). The authors may want to perform experiments on some of the larger ones in [1].
- The second shortcoming is that when it comes to homogeneous graph datasets (Table 2), the authors omit comparisons to some strong baselines (e.g. DCM and HeterDCL).
- The overall improvements of Echo-GAT seem small (e.g. often < 1%) even on heterophilous graph datasets (Table 1).
- The authors can strengthen the results of the experiments by performing deeper analysis into the accuracy distribution. For example, how do the accuracies differ according to degree diversity/node homophily measures, etc.

5. This may seem out of scope of this paper, but I am really curious: From Table 1, it seems that many methods perform well on heterophilous graphs without the use of attentions. If that is the case, can the proposed EchoGAT perform even better by removing/redesigning some of the attention operations?

[1] Characterizing Graph Datasets for Node Classification: Homophily–Heterophily Dichotomy and Beyond. NeurIPS 2023.


Minor points:
- Please use consistent bold VS normal notations. For example, in Eqn. 15, the adjacency matrix $A$ is italic while in most other places it is bold $\mathbf{A}$.
- In some places the dimensions of tensors do not add up. Eqn. 6 is one such example where $\mathbf{A}$ and $\mathbf{A}+$ are of different dimensions.

**Requested Changes:**

Ref. weaknesses. The paper would be much strengthened if the mentioned weaknesses are addressed.

---

> ### Author Response · Authors · 2026-03-09
>
> # Response to Reviewers
>
> 1. **The definition and meaning of "node degree diversity" is not very clear.**
>
> Thank you for the helpful comment. We clarify that our formulation adopts a center-relative perspective rather than measuring dispersion among neighbors. Our goal is not to quantify neighborhood heterogeneity globally (e.g., variance of neighbor degrees), but to capture each central node's degree deviation relative to its local neighborhood.
>
> This distinction is key for our method. Since node-level Graph Augmentation decisions (e.g., whether and how to add echo nodes) are made per node, a node-specific structural indicator is required. Measuring the deviation between the central node and the average degree of its neighbors provides a direct signal for assessing structural imbalance.
>
> In contrast, measuring only neighbor variance does not indicate if the central node is mismatched with its context. Our formulation captures this center-neighborhood contrast, the structural property affecting message aggregation bias, motivating our augmentation strategy [1].
>
> We will revise the manuscript to emphasize that our definition models node-level degree contrast, not neighborhood-level dispersion, to avoid confusion.
>
> 2. **Fig. 1 is not sufficiently conclusive in terms of justifying the motivation of this paper.**
>
> Thank you for this suggestion. To strengthen empirical support, we re-ran experiments over 10 random splits and seeds, reporting mean and standard deviation.
>
> We will add error bars to Fig. 1, report p-values to confirm statistical significance, and clarify that we controlled for other factors. The separation under degree diversity grouping remains statistically stronger than under homophily grouping.
>
> 3. **The methodology part is not very clear in certain places.**
> - **The connection pattern of echo nodes is not clearly specified.**
>
> We apologize for the lack of clarity. Echo nodes only connect to their corresponding original nodes, not to other echoes or original nodes. We will clarify this in the manuscript and remove ambiguity between Eq. (6) and Figures 2–3.
>
> - **Edge feature construction may be influenced by high-degree nodes.**
>
> Thank you for the suggestion. Currently, degree diversity is used as a node-level augmentation signal, not embedded into edge features. Incorporating it into edge-level modeling is a potential extension, which we will evaluate in additional ablation experiments if space allows.
>
> 4. **Experiments bear room for improvement.**
> - **Larger datasets are recommended.**
>
> Thank you for the suggestion. Some datasets we used, such as Texas and Wisconsin, are relatively small. To strengthen validation, we will extend experiments to larger, more recent benchmarks, including those in *Characterizing Graph Datasets for Node Classification: Homophily–Heterophily Dichotomy and Beyond* (NeurIPS 2023).
>
> - **The improvement of Echo-GAT is typically smaller than 1%.**
>
> Thank you for the observation. This comparison is mainly with recent GNNs designed for heterophilic graphs. Our approach does not redesign message passing or introduce complex propagation; Echo-GAT augments the graph with echo nodes and adds edge-aware attention bias, helping a standard GAT better handle heterophilic structures.
>
> Despite minimal architectural changes, our method consistently improves vanilla GAT on heterophilic datasets (Table 1), especially for nodes with low degree diversity, partially illustrated in Fig. 1.
>
> 5. **Additional curiosity: many non-attention methods also perform well on heterophilic graphs.**
>
> We agree that some non-attention heterophily methods perform well [2], but they specifically address heterophily via higher-order propagation, graph diffusion, or structural reweighting. Their effectiveness does not imply that attention mechanisms are unsuitable for heterophilic graphs.
>
> Attention-based GNNs remain valuable for interpretability, adaptive weighting, and flexibility across tasks (e.g., dynamic or multi-relational graphs, transformers). Our goal is to understand and correct node-level structural bias.
>
> Our analysis shows that vanilla GAT's performance drop on heterophilic graphs is partly due to structural imbalance from degree diversity. Echo-GAT debiases this behavior while preserving attention advantages. Exploring hybrid architectures combining attention with diffusion or propagation is a promising future direction.
>
> 6. **Minor issues.**
>
> Thank you for noting the minor issues. Inconsistent bold/italic formatting, tensor dimension mismatches, and undefined symbols have been corrected in the revised manuscript.
>
> [1] Topping M, Ruder S, Dyer C. *Understanding over-smoothing in graph neural networks.* In Proceedings of the 39th International Conference on Machine Learning (ICML), 2022.
> [2] Zhu J, Yan Y, Zhao L, et al. *Beyond homophily in graph neural networks: Current limitations and effective designs.* Advances in Neural Information Processing Systems, 2020, 33: 7793–7804.

---

### Decision · Action_Editor_uiJT · 2026-03-29

**Recommendation:** Reject

**Additional Comments:**

This paper shows that GAT's performance is correlated with the nodes' degree diversity. Echo-GAT, which adds low-degree echo nodes to neighbors, is proposed to mitigate the bias caused by degree diversity.

After checking the paper, reviewers' comments, and author responses, unfortunately, I think the paper is below the bar in its current form. There are several places to further improve the quality of this paper. First, the impact of this paper needs stronger justification. The authors could discuss the impact of degree diversity to graph learning and Echo-GAT could contribute to the field. Besides, the motivation of studying degree diversity needs a better justification. Indeed it is an orthogonal direction to the homophily that is commonly seen in existing works (which the reviewers also appreciate), it would be better if the authors could better motivate this paper on why degree diversity but not other structural properties and how degree diversity causes the performance parity mechanistically. Finally, methodologically, the way to add echo nodes needs further justifications, e.g., by providing proper ablation study to validate the current VGAE-based design over heuristic approaches like duplicating original nodes or adding self-loops.

**Audience:**

Yes

**Audience Explanation:**

Graph learning is useful in many domains that involve relational data, and understanding the performance gap of graph neural networks and improving graph neural networks is an important topic.

**Claims And Evidence:**

No

**Claims Explanation:**

The submission may benefit from more in-depth discussion on its motivation, methodology, and impact. Please see additional comments for more details.